EMBO
Molecular Medicine

# *HSD17B7* gene in self-renewal and oncogenicity of keratinocytes from Black versus White populations

Xiaoying Xu[1,†] (iD), Beatrice Tassone[1,†] (iD), Paola Ostano[2] (iD), Atul Katarkar[1] (iD), Tatiana Proust[1] (iD), Jean-Marc Joseph[3] (iD), Chiara Riganti[4] (iD), Giovanna Chiorino[2] (iD), Zoltan Kutalik[5] (iD), Karine Lefort[1,‡] (iD) & Gian Paolo Dotto[1,6,7,*] (iD)

## Abstract

**Human populations of Black African ancestry have a relatively high risk of aggressive cancer types, including keratinocyte-derived squamous cell carcinomas (SCCs). We show that primary keratinocytes (HKCs) from Black African (Black) versus White Caucasian (White) individuals have on average higher oncogenic and self-renewal potential, which are inversely related to mitochondrial electron transfer chain activity and ATP and ROS production. *HSD17B7* is the top-ranked differentially expressed gene in HKCs and Head/Neck SCCs from individuals of Black African versus Caucasian ancestries, with several ancestry-specific eQTLs linked to its expression. Mirroring the differences between Black and White HKCs, modulation of the gene, coding for an enzyme involved in sex steroid and cholesterol biosynthesis, determines HKC and SCC cell proliferation and oncogenicity as well as mitochondrial OXPHOS activity. Overall, the findings point to a targetable determinant of cancer susceptibility among different human populations, amenable to prevention and management of the disease.**

**Keywords** genetic cancer susceptibility; HSD enzymes; OXPHOS; squamous cell carcinoma; stem cell potential
**Subject Categories** Cancer; Skin

## Introduction

Human populations with different genetic backgrounds differ significantly in susceptibility to many cancer types (Ozdemir & Dotto, 2017; Li *et al*, 2020; Zavala *et al*, 2021). These differences can result from a combination of socioeconomical and behavioral factors, but disparities in cancer risk persist after normalization for these factors and in equal access settings, such as in the US Military Health System (Andaya *et al*, 2013; Schreiber *et al*, 2014; Dess *et al*, 2019). In investigating this question, we will be employing the term "*ancestry*" in reference to individuals with common genetic and phenotypic features rather than "*race*", given the complex and problematic use of the latter term (Cooper, 2013). Relative to other populations, individuals of Black African ancestry have an increased risk of various types of aggressive cancer, including keratinocyte-derived squamous cell carcinomas (SCCs) that arise at various body sites (Van Loon *et al*, 2018; Olusola *et al*, 2019; Nocon *et al*, 2020). Even skin SCCs, although having a markedly lower incidence most likely linked to pigment photoprotection, exhibit a more aggressive behavior in patients of Black African background, with frequent metastasis (Agbai *et al*, 2014).

Together with a variety of socioeconomic factors, differences in cancer susceptibility can be attributed, in part, to population-specific genetic and epigenetic determinants of organismic functions such as metabolism, native and acquired immunity, and drug detoxification (Ozdemir & Dotto, 2017; Li *et al*, 2020; Zavala *et al*, 2021). Only limited evidence exists for differences in cancer stem-like cells (CSCs) of origin, in spite of the key role that these cells can have in the disease, with implications mostly based on marker association studies (Farhana *et al*, 2016; Wang *et al*, 2017; Jiagge *et al*, 2018). Whether or not there are ancestry-related differences of functional significance in normal stem cell populations from which tumors arise has not, to our knowledge, been investigated.

The skin provides a unique resource for relatively large-scale studies of normal and cancer tissues, as these are routinely discarded at the end of surgical procedures. Growth/differentiation and self-renewal properties of skin keratinocytes are shared with

1 Department of Biochemistry, University of Lausanne, Epalinges, Switzerland
2 Cancer Genomics Laboratory, Fondazione Edo ed Elvo Tempia, Biella, Italy
3 Division of Pediatric Surgery, Women-Mother-Child Department, Lausanne University Hospital (CHUV), Lausanne, Switzerland
4 Department of Oncology, University of Turin, Turin, Italy
5 University Center for Primary Care and Public Health, University of Lausanne, Lausanne, Switzerland
6 Cutaneous Biology Research Center, Massachusetts General Hospital, Charlestown, MA, USA
7 International Cancer Prevention Institute, Epalinges, Switzerland
*Corresponding author. Tel: +41 21 6925714; E-mail: paolo.dotto@unil.ch
†These authors contributed equally to this work
‡Present address: Department of Laboratory Medicine and Pathology, Institute of Pathology, Lausanne University Hospital and Lausanne University, Lausanne, Switzerland

those of internal stratified epithelial tissues, such as oropharyngeal and laryngeal cavities and esophagus. Squamous cell carcinomas (SCCs) that arise from all these tissues have also common genetic and epigenetic alterations of functional significance (Dotto & Rustgi, 2016). Compromised squamous differentiation underlies keratinocyte to SCC cell transformation (Dotto & Rustgi, 2016). As for other cellular systems (Ito et al, 2006; Tothova et al, 2007; Owusu-Ansah & Banerjee, 2009; Tormos et al, 2011), even in keratinocytes increased oxidative phosphorylation (OXPHOS) and mitochondrial ROS production are involved in promoting commitment of stem cells toward differentiation (Hamanaka et al, 2013; Bhaduri et al, 2015).

Here, we show that foreskin-derived primary human keratinocytes (HKCs) from individuals of Black African ancestries have on average higher oncogenic and stem cell potential than HKCs from Caucasian individuals. Underlying these findings, HKCs from Black African versus Caucasian individuals exhibit lower mitochondrial electron transfer activity as well as ATP and ROS production. These differences are reproduced by modulation of the *HSD17B7* gene, coding for a member of the hydroxysteroid dehydrogenase family of enzymes with a role in sex steroid and cholesterol biosynthesis (Saloniemi et al, 2012) and estradiol function in breast cancer (Shehu et al, 2011; Hilborn et al, 2017), but a so far unsuspected role in control of stem cell potential and OXPHOS activity.

## Results

### Primary keratinocytes from individuals of Black African versus Caucasian ancestry exhibit higher oncogenic and self-renewal potential

We postulated that the greater susceptibility of Black African (Black) versus Caucasian (White) populations to SCCs (Van Loon et al, 2018; Olusola et al, 2019; Nocon et al, 2020) is associated with a different oncogenic and self-renewal potential of their cells of origin. To test this hypothesis, we undertook a combination of functional and genomic/biochemical approaches.

A collection of foreskin-derived primary keratinocyte strains (HKCs) was established from a cohort of young boys of Black African versus Caucasian origin. DNA samples were analyzed by SNP array hybridization (Infinium HumanOmni 2.5-8 arrays, containing approximately 2.5 million human SNPs). Principal component analysis (PCA) of the SNP profiles showed excellent correspondence of genetic profiles with skin phototypes and patients' "self-reported" origins. Samples from individuals of White descent and skin phototypes 1 and 2 clustered tightly in one PCA group, while those of Black descent and skin phototypes 5 and 6 clustered separately, with three subgroups matching geographic distributions of "self–reported" origin within the African continent (Fig 1A). Admixture analysis of the SNP genotype datasets (Alexander et al, 2009) was used to estimate genetic relatedness of donors, clustering them according to an increasing number of possible ancestral populations. With the simplest assumption of two ancestries ($K = 2$), the genome of individuals of self-reported Black origin was found to harbor various levels of the White ancestry genome (Fig 1B).

Oncogenic conversion of human primary keratinocytes (HKCs) can be effectively induced by the concomitant expression of dominant-negative *TP53* ($TP53^{R248W}$) and activated *H-RAS* (*H-RAS*$^{G12V}$) mutant genes (Khavari, 2006). Early passage foreskin HKCs from multiple Black and White individuals were infected with lentiviruses expressing the two genes and tested for their behavior in an orthotopic model of skin SCC development, based on intradermal injection of cells in immune-compromised NOD/SCID mice (Al Labban et al, 2018). We performed 7 independent assays with multiple HKC strains, 4-5 mice per assay. As shown in Fig 2A and B (and Appendix Fig S1A and B), ΔN-p53 and activated *H-RAS* expressing HKCs from Black donors grew into larger tumors with higher tumor cell density than corresponding HKCs from White donors. There were statistically significant differences in tumor formation between HKCs of White and Black individuals, taking the latter as a total group or one with < 20% genomic admixture (Fig 2A). Increased tumor formation was associated with increased cellular proliferation as assessed by Ki67 positivity and decreased squamous differentiation as assessed by keratin 1 and 10 markers expression (Fig 2C and D, Appendix Fig S1C and D).

The greater oncogenicity of HKCs from Black African versus Caucasian individuals could be linked to differences in self-renewal potential. This can be readily assessed by *in vitro* clonogenicity assays, with cells with greater proliferative potential giving rise to large expanding colonies ("holoclones") while those with more restricted dividing capabilities producing colonies of smaller size (Barrandon & Green, 1987). Use of these assays made it possible to significantly increase the number of tested HKC strains. HKCs from a cohort of Black individuals showed on average a greater holoclone-forming capability than those from a White cohort (Fig 2E and Appendix Fig S2A). 3D-sphere formation assays provide an alternative measure of the self-renewal potential of various cell types, including keratinocytes and SCC cells (Al Labban et al, 2018). Similar to the clonogenicity assays, HKCs from Black donors formed in general spheres of larger size than those from White individuals (Fig 2F). Differences in clonogenicity and sphere-forming capability of HKCs from White versus Black individuals, as a total group or one with < or > 20% genomic admixture, were all statistically significant (Fig 2E and F).

Thus, HKCs from individuals of Black versus White ancestries have on average greater oncogenic and proliferation potential.

### Keratinocytes from individuals of Black versus White ancestries have a differential OXPHOS-associated gene signature

To probe into underlying mechanisms, we undertook a global transcriptomic analysis of HKC strains from individuals of the two populations. By comparing the profiles of HKCs from individuals of the two groups, a set of 164 genes (as characterized transcripts) were, on average, differentially expressed in HKCs from Black versus White donors ($\log_2$(Foldchange) > 0.58 and $P$ value < 0.05), with an overall good agreement with admixture genotypes (Fig 3A and Dataset EV1). Clustering of the signature genes by the Manhattan distance metrics method (Shirkhorshidi et al, 2015) identified four main groups of co-expressed genes (Fig 3A), with gene ontology (GO) analysis showing highly significant enrichment of gene families involved in various metabolic and detoxification processes, with similar or identical functional categories when HKCs from Black individuals were considered as a total group or one with < 20% genomic admixture (Fig 3B and Dataset EV1).

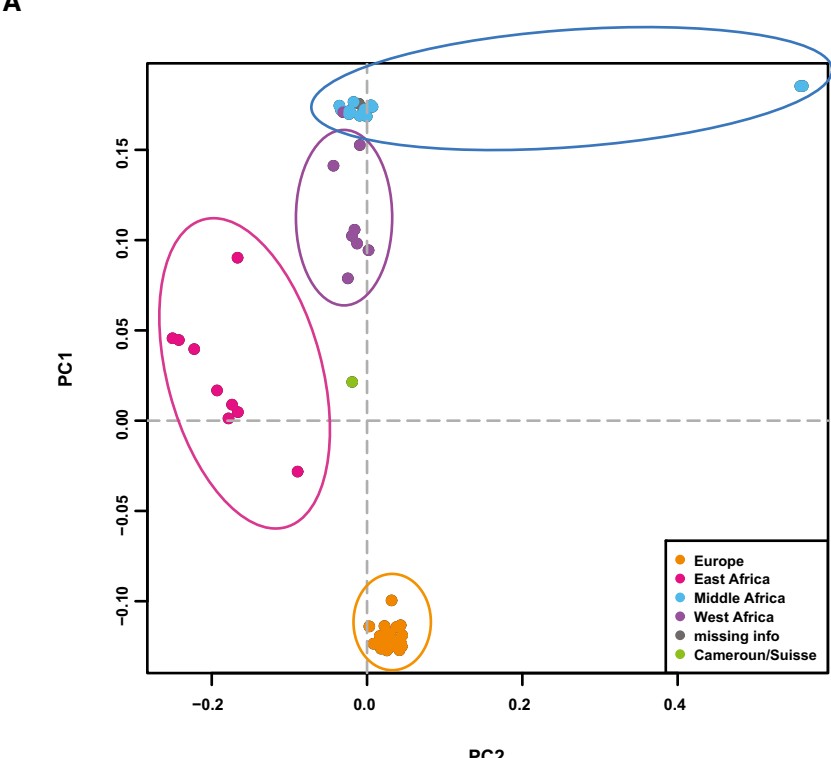

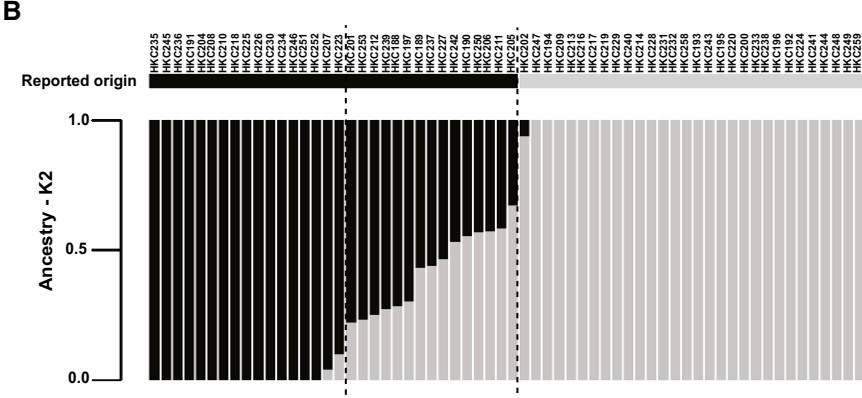

**Figure 1. Distinct genomic profiles of cells from individuals of Back versus White ancestries are accompanied by signs of genomic admixture and variation.**

A Principal component analysis (PCA) based on 53,837 single nucleotide polymorphism (SNPs) markers of 62 foreskin donors assessed by microarray chip hybridization (via Illumina Human Omni2.5). PLINK (Package: PLINK1.07) and GCTA analysis tools were used for genetic distance analysis. The first two components (PC1 and PC2) are plotted and explain 5.1% of the variance. Dotted line corresponds to eigenvectors at 0. Each point represents a foreskin donor. Points are labeled and grouped according to self-reported donors' country of origin.

B Admixture analysis of the indicated HKC strains of Black versus White individuals, on the basis of the SNP genotype dataset with the simplest assumption of two ancestries ($K = 2$). On the y-axis admixture proportions are represented as gray and black bars. On the x-axis, individual samples are reported. Dashed lines delimit Black individuals with more than 20% admixture of the other ancestry.

While the above analysis was based on the identified set of differentially expressed genes, for further insights we resorted to Gene Set Enrichment Analysis (GSEA) of the total gene expression profiles of HKC strains from Black versus White individuals. Out of a hallmark of 50 gene signatures from the Molecular Signature Database (MSigDB 50 hallmarks: https://www.gsea-msigdb.org/gsea/msigdb), we found a significant enrichment (FDR < 0.05)

for a mitochondrial oxidative phosphorylation (OXPHOS) gene signature, with no significant enrichment for other signatures (Fig 3C).

Thus, HKC strains from Black versus White individuals can be distinguished—on average—on the basis of genotype, a number of genes enriched for metabolic detoxifying functions and, more globally, an OXPHOS-related gene signature.

### HSD17B7 is a differentially expressed gene in Black versus White HKCs of prognostic significance

To identify individual genes of functional significance, we compared transcriptomic profiles of HKCs from Black versus White individuals with those of a large data set of Head/Neck SCCs (HNSCCs) from patients of the two ancestries (520 patients, 452 White, 48 Black;

TCGA Firehose Legacy, November 2020, from cBioportal (Gao *et al*, 2013)). Twenty-one genes were found to be differentially expressed in both data sets of HKCs and HNSCCs from Black versus White individuals (Fig 4A). Many of them are involved in metabolic or detoxifying functions, including several *GSTM* family members, coding for glutathione S transferases (GST) with a possible complex role in cancer development (Chatterjee & Gupta, 2018).

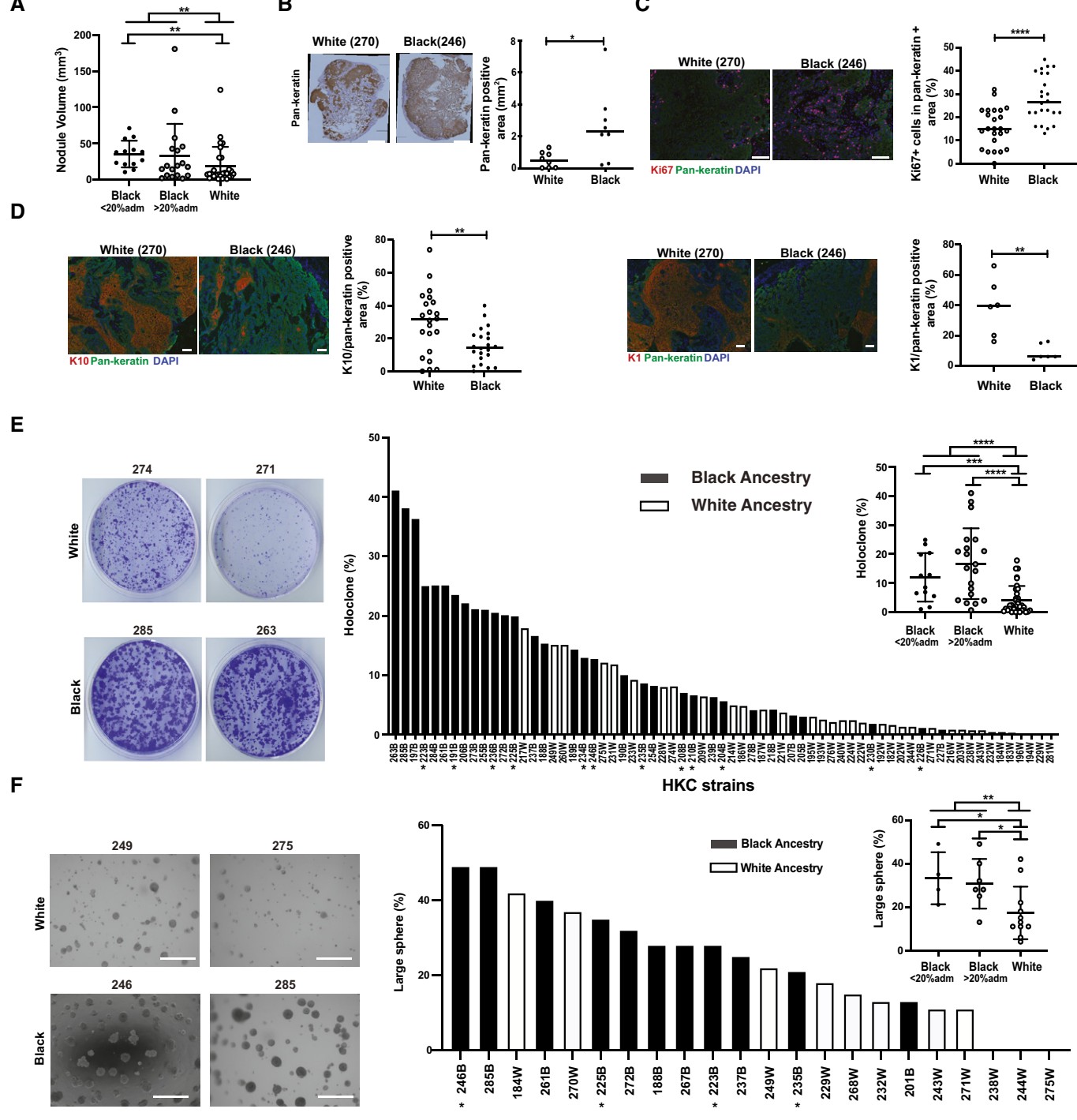

**Figure 2.**

**Figure 2. HKCs from individuals of Back versus White descent display greater oncogenic and self-renewal potential.**

A  Multiple HKC strains (passage 2,3) from individuals of the two ancestries were stably infected with a lentivirus expressing dominant-negative mutant *TP53* (*TP53^R248W*) and superinfected with a *H-Ras^V12* expressing virus, followed, 24 h later, by parallel intradermal injections into the back of NOD-SCID female mice. 7 strains per ancestry, 4–5 mice per assay were used. Tumors were excised 15 days post-injection. Shown is size quantification of all individual tumors (dots) by digital caliper. **P < 0.01. n(tumors by White HKCs) = 28, n(tumors by Black HKCs) = 31, n(tumors by Black HKCs with < 20% genome admixture) = 13. Two-tailed unpaired Mann–Whitney test. In the plot, bars represent mean ± SD.

B  Representative immunohistochemical analysis with anti-pan-keratin antibodies of a pair of tumor lesions formed by parallel injections of White versus Black HKC strains together with quantification of the pan-keratin positive areas (mm²) of individual lesions (dots) formed by 6 HKCs strains of the two ancestries. n(tumors per ancestry) = 8. *P < 0.05. Horizontal line: median. Two-tailed unpaired *t*-test. Scale bar: 500 μm.

C, D  Double immunofluorescence analysis of tumors formed by 6 HKC strains, as in the previous panels, with antibodies against Ki67 (C) and keratin 10 and 1 (K10, K1) (D) together with anti-pan-keratin antibody for tumor cell identification. Shown are representative images and quantification of results of individual lesions (dots). Digitally acquired images were used to determine the % of Ki67-positive cells (C) and of K1/K10 positivity (D) in pan-keratin-positive areas, examining 3-5 fields in each case. For Ki67, K10, and K1 staining: n(tumors per ancestry) = 24, 22, and 6, respectively. ****P < 0.0001; **P < 0.005, horizontal line: median, two-tailed unpaired *t*-test. Scale bars: 100 μm.

E  Colony-forming assay of multiple HKCs from White versus Black donors (passages 2,3) plated in triplicate dishes, at limited density (1,000 cells per 60-mm dish). At day 8, cells were fixed and stained by crystal violet, and percentage of large colonies ("*holoclones*"), 10x size relative to total number of colonies in the dish, was determined by Fiji software. Shown are representative images and bar graph quantification in decreasing order of holoclone-forming capability by all indicated HKC strains (three dishes per strain). Asterisks indicate Black strains with < 20% admixture (Fig 1B). Inset: pooled values of all HKCs from White versus Black ancestries shown as individual dots together with mean ± SD. n(White HKC strains) = 35; n(Black HKCs, total, < and > 20% genome admixture strains) = 32, 12, and 20, respectively. ***P < 0.0005; ****P < 0.0001, two-tailed unpaired *t*-test.

F  Sphere-forming assays of HKC strains from White and Black donors plated in Matrigel suspension (2,000 cells per 8-well chamber slide, 2–3 wells per strain) and cultured for 8 days. Quantification of large spheres (> 2,000 pixels ≥ 0.0095 mm²) versus total number of spheres (> 100 pixels ≥ 0.00047 mm²) formed by each strain was determined by Fiji software. Shown are representative images together with bar graph quantification in decreasing order of large sphere-forming capability by all indicated HKC strains. Asterisks indicate Black strains with < 20% admixture (Fig 1B) Inset: pooled values of all HKCs from White versus Black ancestries shown as individual dots together with mean ± SD. n(White HKC strains) = 11; n(Black HKCs, total, < and > 20% genome admixture strains) = 11, 4, and 7. **P < 0.01; *P < 0.05. Two-tailed unpaired *t*-test. Scale bars: 300 μm.

*HSD17B7*, coding for an enzyme of the hydroxysteroid 17-beta dehydrogenase family involved in steroid hormones and cholesterol biosynthesis (Saloniemi *et al*, 2012), was the top-ranked gene on the basis of statistical significance for higher expression in both Black versus White HKCs and HNSCCs (Fig 4A and Table EV1). For further functional analyses, we focused on this gene, as examination of HNSCCs patients' clinical history showed a strong positive association between its expression and poor survival. Kaplan–Meier curves showed *HSD17B7*-associated differences for both male and female patients (Fig 4B). As a second approach, multiple-variable Cox regression analysis was used to adjust for patients' sex, age, and ancestry, or all three variables together, finding that in all cases elevated *HSD17B7* expression remained significantly associated with poor patients' survival (Fig 4C).

Correlation analysis of transcriptomic profiles showed a significant association of *HSD17B7* expression, in both HKC strains and HNSCCs, with genes enriched for families related to cellular metabolic processes and mitochondrial compartments as well as DNA repair/replication and the nucleus (Fig 4D). *HSD17B7* expression across all tested HKC strains also correlated with clonogenic potential (Fig 4E).

Differential expression of *HSD17B7* was validated in HKC strains from Black versus White donors by RT-qPCR (Fig 4F) as well as immunofluorescence, the latter showing a particulate perinuclear distribution of the enzyme non-overlapping with mitochondria (Fig 4G and Appendix Fig S3A–C), consistent with its previously reported subcellular localization (Marijanovic *et al*, 2003).

Expression quantitative trait loci (eQTLs) can provide an explanation for differentially regulated genes between ancestries. By integrated analysis of SNPs and transcriptomic profiles, we identified 9 eQTLs within a 1Mb genomic region of the *HSD17B7* gene that were strongly associated with its expression (FDR < 0.005) (Fig 4H and

**Figure 3. Transcriptomic profiles of Black versus White HKCs are characterized by an OXPHOS-related gene signature.**

A  Top bar: gradient of population admixture of the indicated HKC strains of Black versus White individuals, on the basis of the SNP genotype analysis shown in Fig 1B, with the simplest assumption of two ancestries (K = 2). Gradient ranges from black to white colors for Black and White individuals with no population admixture, respectively. Individuals with increasing proportions of White population admixture are represented with degrading shades of gray. Bottom: Heatmap of differentially expressed genes (characterized transcripts) in HKC strains from Black versus White individuals ordered according to genome admixture calculations. 164 genes with significantly different expression levels between the two groups (log₂(Foldchange)|> 0.58 and P value < 0.05) were identified (Dataset EV1). Modified Z scores of the individual genes, as median-centered log2 intensity values divided by standard deviation, are shown by color gradient variations. Unsupervised hierarchical clustering of genes was based on their modified Z scores, using Manhattan distance as distance metric and complete linkage method (Shirkhorshidi *et al*, 2015). Four main groups of co-expressed genes were identified (purple triangles).

B  Gene Ontology (GO) analysis of the differentially expressed genes identified in the previous panel. Shown is a list of process networks with statistical significance of enrichment in Black HKCs, total or with < 20% genomic admixture (upper and lower panels, respectively). The complete list is provided in Dataset EV1. Fisher's exact *P*-values are reported.

C  Gene set enrichment analysis of the global expression profiles of the Black versus White HKC strains (GSE156011) using 50 HALLMARK signatures (https://www.gsea-msigdb.org/gsea/msigdb/collections.jsp#H), revealing a significant association with the OXPHOS signature. Genes are ranked by signal-to-noise ratio based on their differential expression in Black versus White HKC strains. Position of genes in each gene set is indicated by black vertical bars and the enrichment score (ES) by the green line. Normalized Enrichment Score (NES), nominal *P*-value, and FDR q-value, based on the permutation-generated null distribution, are as shown.

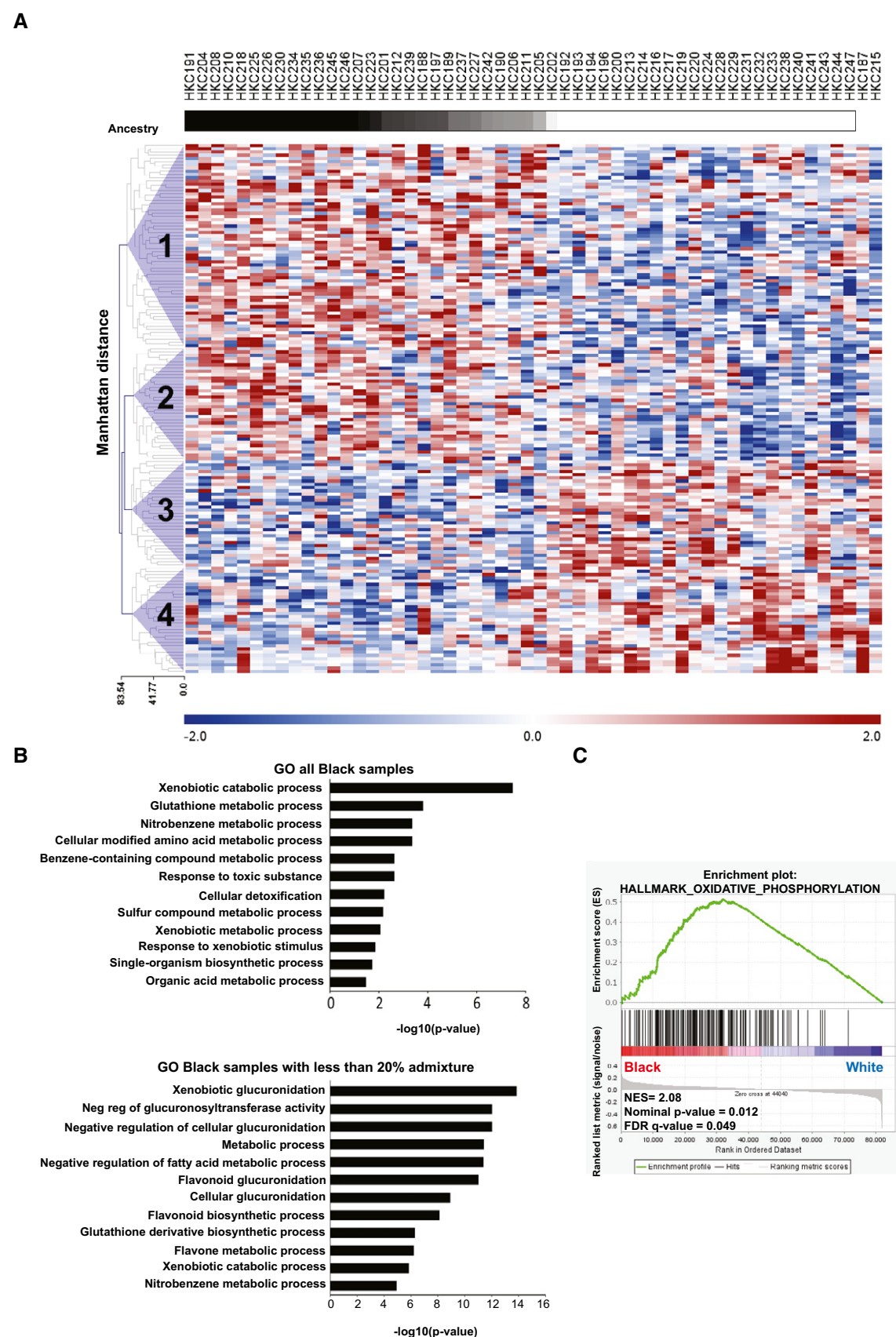

Figure 3.

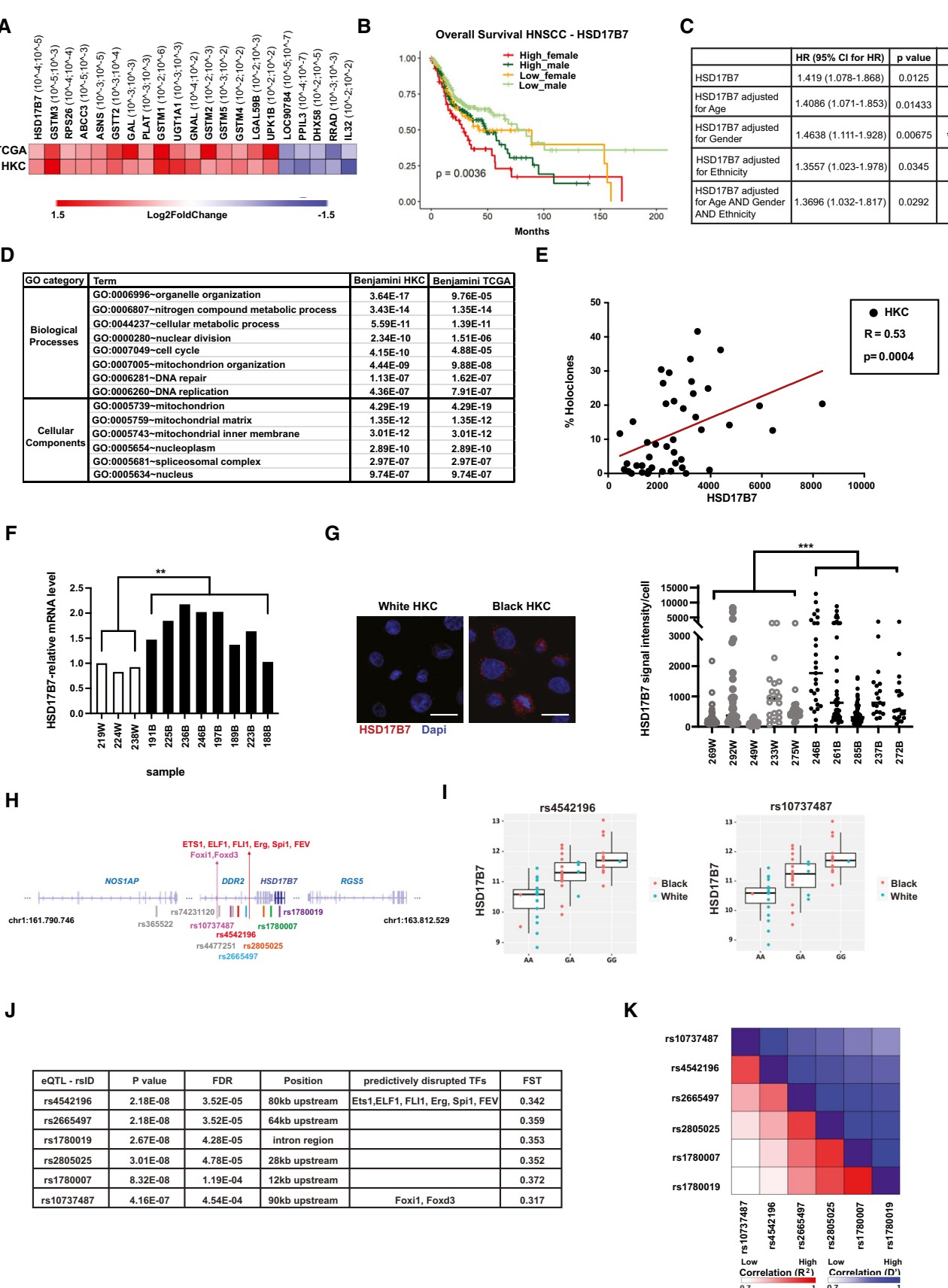

**Figure 4.**

**Figure 4. *HSD17B7* is a differentially expressed gene in Black versus White HKCs and HNSCCs of prognostic significance.**

A   Differentially expressed genes (DEGs) in Black versus White HKC strains identified in the present study were compared with those in a publicly available data set of HNSCCs (TCGA Firehose Legacy, 520 patients) selected on the basis of the same criteria (|log$_2$(Foldchange)|> 0.5 and *P* value < 0.05) as for analysis of the HKC profiles in Table S2. Shown is the list of the overlapping DEGs together with a heatmap indication of average log2 fold difference and corresponding statistical significance in both HKC and TCGA (HNSCC) data sets (within brackets).

B   Kaplan–Meier plots of long-term overall survival of HNSCC patients from the TCGA dataset divided on the basis of high versus low *HSD17B7* expression levels relative to its median expression value among males (darkgreen and lightgreen lines) and females (red and orange lines), showing statistically significant association of elevated HSD17B7 expression with poorer patient survival. Statistical significance was calculated by the log-rank test between the groups. High_female: *n* = 57 patients. Low_female: *n* = 79 patients. High_male: *n* = 203 patients. Low_male: *n* = 181 patients.

C   Single and multiple COX regression analysis of HNSCC data sets from the TCGA for high risk of patients' death as a function of levels of *HSD17B7* expression, either as a single variable or after adjusting for gender, ethnicity, age, or all three together. Only HNSCC samples from patients of White (Caucasian) versus Black ancestries were used for this analysis. Cox proportional hazards regression model was fitted with the R function coxph.

D   Gene Ontology (GO) analysis of the list of genes that positively correlate with *HSD17B7* expression in HKC strains and HNSCCs, with corresponding Benjamini scores (left and right columns, respectively). GO categories are divided into Biological Processes and Cellular Components and the most significant categories in both HKC strains and TCGA (HNSCC) are reported.

E   Correlation analysis between the holoclone-forming capability of HKC strains (as determined in Fig 2E) and *HSD17B7* expression levels in the corresponding transcriptomic profiles. Values for individual HKC strains are indicated by dots. *n*(HKC strains) = 44. The linear regression R (0.53) and *P*-value (0.0004) are indicated. The graph shows absolute expression values (probe intensity after global normalization by TAC software) obtained from human Clariom D microarrays. The statistical test is a simple linear regression.

F   RT-qPCR analysis of *HSD17B7* expression in the indicated HKC strains derived from White versus Black donors (white and black bars), as a validation of the corresponding transcriptomic profiles of Fig 3A.**P* < 0.01.

G   Immunofluorescence (IF) analysis of HSD17B7 protein expression in White and Black HKCs. Shown are representative images as well as IF signal quantification at the level of individual cells (dots) for the indicated strains. 19–46 cells were counted in each case. *n*(HKC strains per ancestry) = 5, ***P* < 0.0005, horizontal line: median, 2-tailed unpaired *t*-test. Unit of y-axis: pixels. Scale bar: 20 μm.

H   Map of eQTL position within a 1 MB region encompassing the *HSD17B7* gene, with corresponding exons shown as navy blue bars, and the additional indicated genes. Shown are all eQTLs associated with HSD17B7 expression in the HKC transcriptomic profiles, with the ones with differential distribution in the Black versus White HKC strains indicated in colors. Position of the eQTLs affecting recognition of the indicated transcription factors is also shown.

I   Plots with allele frequency and associated levels of *HSD17B7* expression in Black versus White samples for two of the identified eQTLs (rs4542196 and rs10737487). Log2 intensity values are reported on the *y*-axis, while genotypes (AA, GA, GG) are reported on the *x*-axis. Rs4542196 (AA: 19 White, 2 Black; GA: 3 White, 14 Black; GG: 1 White, 12 Black). Rs10737487 (AA: 18 White, 1 Black; GA: 5 White, 15 Black; GG: 1 White, 12 Black). Boxes represent the first (lower) quartile, the median (central band) quartile, and the third (upper) quartile. Whiskers extend to the most extreme data point which is no more than 1.5 times the interquartile range from the box.

J   Summary table of eQTLs with statistically significant association with *HSD17B7* expression in experimentally determined HKC transcription profiles (52 strains) and differential distribution in Black versus White populations by analysis of 1,000 Genomes Phase I May 2011 [AFRICA (*N* = 246), EUROPE (*N* = 380)] with Fixation index ($F_{ST}$) value > 0.3. The effect of the genotype was modeled as additive linear and tested for its significance (*P*-values) using t-statistic.

K   Linkage analysis on the six reported SNPs using the LDmatrix tool (https://ldlink.nci.nih.gov/?tab=ldmatrix). Pairwise linkage disequilibrium statistics, calculated on all populations (AFR, AMR, EAS, EUR, SAS), are reported as heat maps of shaded colors corresponding to the indicated gradient of values (red: $R^2$; blue: D′). The specific $R^2$ and D′ numbers are provided in Appendix Table S3.

Dataset EV2). Combined allele frequency analysis of our SNP data sets and the 1,000 genome phase I project (probed by SPSmart online software Amigo *et al*, 2008; Santos *et al*, 2016) showed that 6 of these eQTLs have significantly different allele frequency in the Black versus White populations (FST > 0.3) (Fig 4H–J), with strong linkage disequilibrium (D′ > 0.5, using the LDMatrix tool (Alexander & Machiela, 2020) (Fig 4K and Dataset EV2). Analysis of the GTEX database showed that the ancestry-specific eQTLs for *HSD17B7* that we have identified are significantly associated with *HSD17B7* expression in multiple tissues including skin and surface epithelia (Table EV2).

Thus, *HSD17B7* is the most significantly highly expressed gene in HKCs and HNSCCs of Black versus White individuals, with specific eQTLs differentially distributed between the two ancestries.

### *HSD17B7* is a determinant of HKC self-renewal and tumorigenic potential

Prompted by the above results, we postulated that *HSD17B7* is involved in self-renewal and tumorigenic potential of HKCs and SCC cells. To assess this possibility, we resorted to lentivirus-mediated overexpression and gene silencing approaches. *HSD17B7* overexpression was sufficient to enhance clonogenicity of a number of HKC strains from White donors (Fig 5A and Appendix Fig S2B).

Like primary HKCs, established cancer cell lines, including SCC cells, also contain distinct subpopulations with different growth/tumorigenic potential (Al Labban *et al*, 2018). As with HKCs, clonogenicity of skin and head/neck SCC cell lines was enhanced by *HSD17B7* overexpression (Fig 5B and Appendix Fig S2C), with an overall increase of their rate of proliferation (Fig 5C).

To assess the *in vivo* significance of the findings, HKCs plus/minus *HSD17B7* overexpression were infected with ΔN-p53 and activated *H-RAS* expressing lentiviruses, followed by intradermal tumorigenicity assays into immune-compromised mice. As shown in Fig 5D–F and Appendix Fig S4A and B, *HSD17B7* overexpressing HKCs gave rise to tumors of greater cell density and proliferative index and lesser differentiation than controls, recapitulating the differences observed in similar assays with HKCs from Black versus White donors (Fig 2A–D). Similar assays were performed with SCC cells plus/minus *HSD17B7* overexpression. Even in this case, *HSD17B7* overexpression resulted in tumors with higher cell density and proliferative index (Fig 5G and H).

Gene knockdown experiments were performed to assess whether *HSD17B7* exerts also a converse permissive function in proliferation. Down-modulation of the gene using two different shRNA silencing lentiviruses (Appendix Fig S5A–C) suppressed effectively clonogenicity of multiple HKC strains irrespective of ancestry, also reducing sphere-forming capability and cell proliferation as assessed by

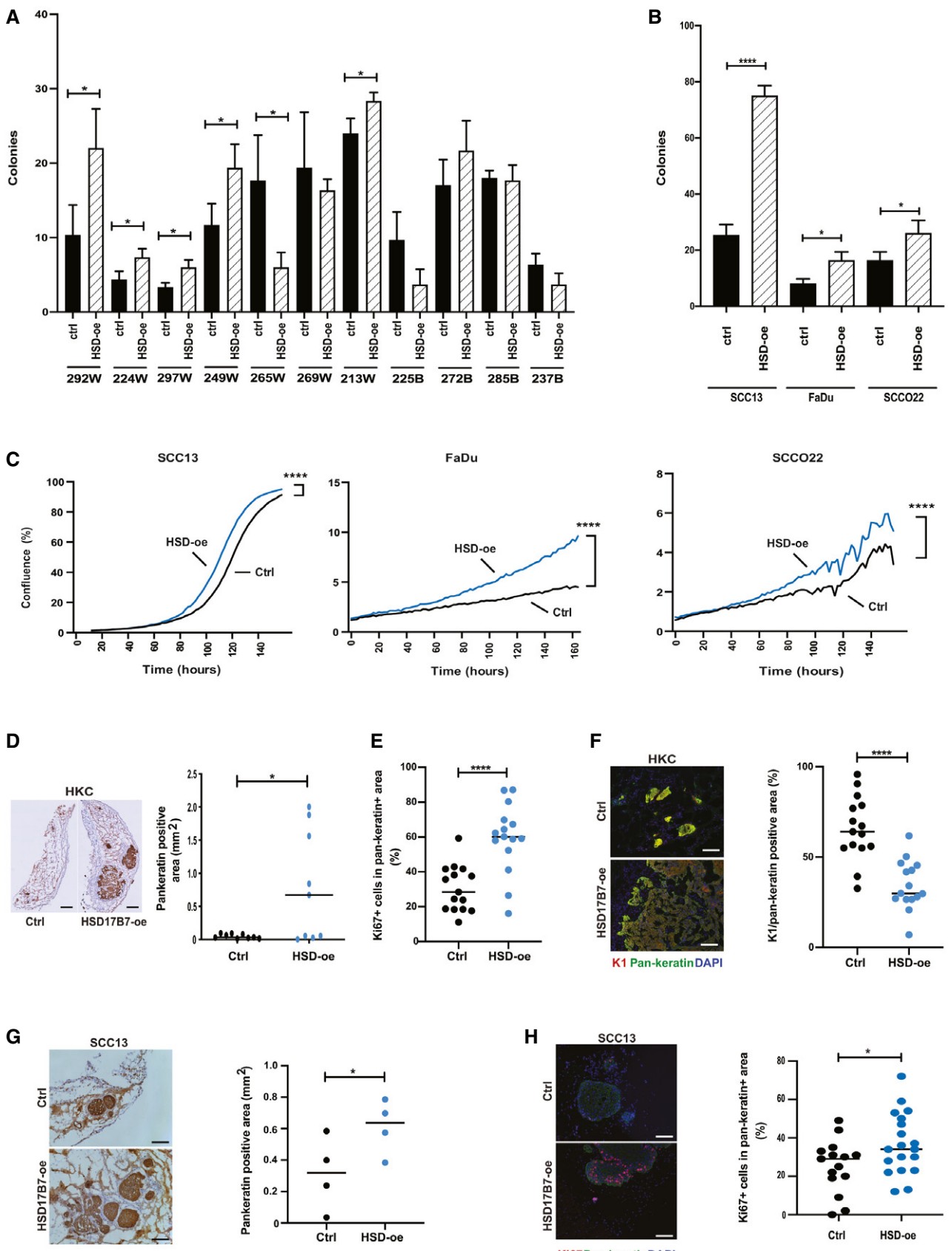

**Figure 5.**

**Figure 5.** *HSD17B7* **is a positive determinant of HKC and SCC cell proliferative and oncogenic potential.**

A   Colony-forming assays of multiple HKC strains from White and Black donors (passages 2,3) infected with an *HSD17B7*-expressing lentivirus (HSD-oe) versus empty vector control (ctrl), plated in triplicate dishes at limited density (1,000 cells per 60-mm dish) at 5 days after infection. A week later cells were fixed and stained by crystal violet, and number of colonies (> 0.149 mm$^2$) was determined. Shown are numbers of colonies ± SD. *n*(dishes per condition) = 3. *$P$ < 0.05. 2-tailed unpaired *t*-test.

B   Colony-forming assays of the indicated SCC cell lines stably infected with an *HSD17B7*-expressing lentivirus (HSD-oe) versus empty vector control (ctrl), utilizing the same conditions as in the previous panel. Shown are numbers of colonies ± SD. *n*(dishes per condition) = 3. *$P$ < 0.05; ****$P$ < 0.0001. Two-tailed unpaired *t*-test.

C   Proliferation live-cell imaging assays of SCC cell lines plus/minus *HSD17B7* overexpression as in the previous panel. Cells were plated in triplicate wells in 96-well plates followed by cell density measurements (IncuCyte), taking four images per well every 2 h for 150–160 h. *n* (wells per condition) = 3. ****$P$ < 0.0001. Pearson r correlation.

D–F   Intradermal tumorigenicity assays with HKCs (269W) stably infected with an *HSD17B7* overexpressing lentivirus (HSD17B7-oe) versus control (Ctrl). Experimental conditions were the same as in Fig 2. Mice were sacrificed 15 days after injection. Shown are representative images and quantification of tumor cell density assessed by immunohistochemical staining with anti-pan-keratin antibodies (D), as well as % of Ki67-positive cells (E) and K1 positivity (F) of pan-keratin positive areas. For (D), *n*(tumors per condition) = 9; for (E and F) *n*(3 fields per tumor, 5 tumors per condition) = 15. *$P$ < 0.05, ****$P$ < 0.0001. Horizontal line: median. 2-tail unpaired *t*-test. Scale bar: 500 μm (D), 100 μm (F).

G, H   Intradermal tumorigenicity assays with SCC13 cells overexpressing *HSD17B7* (HSD17B7-oe) versus control (Ctrl). Mice were sacrificed 15 days after injection. Shown are representative images and quantification of tumor cells density, assessed by immunohistochemical staining with anti-pan-keratin antibodies (G) as well as % of Ki67-positive cells in pan-keratin-positive areas (H). For (G), *n*(tumors per condition) = 4; for (H), *n*(3–5 fields per tumor, 4 tumors per condition) 15, 19. *$P$ < 0.05; horizontal line: median. One-tailed paired *t*-test. Scale bar in (G, H): 100 μm.

EdU labeling (Fig 6A–C). *HSD17B7* gene silencing in a panel of SCC cell lines resulted in a similar suppression of clonogenicity and cell proliferation (Fig 6D and E), with no such effects occurring in cells infected with an *HSD17B7* overexpressing lentivirus (Fig 6F).

Thus, as could be predicted from differences between Black versus White HKCs and HNSCCs, modulation of *HSD17B7* expression impacts self-renewal and tumorigenicity of HKCs and SCC cells.

### Differential mitochondrial electron transport chain, ATP, and ROS production in HKCs from Black versus White donors and as a function of *HSD17B7* expression

Previous work showed that elevated OXPHOS and resulting increase in mitochondrial ROS production decrease the life span of cells (Balaban *et al*, 2005) and promote stem cell commitment toward differentiation in a variety of systems, including keratinocytes (Hamanaka *et al*, 2013; Bhaduri *et al*, 2015). Expression profiles of Black African versus Caucasian HKCs were distinguished by an OXPHOS-related gene signature (Fig 3C). Many genes of the signature related to cellular respiration, electron transport chain, and mitochondrial organization and biogenesis were more highly expressed in Black versus White HKCs and up-regulated in three strains of White HKCs by *HSD17B7* overexpression (Fig 7A and B and Dataset EV1).

Expression of the above genes may be inversely related to levels of mitochondrial activity as a compensatory mechanism as reported for mitochondria disorders and metabolic conditions resulting in OXPHOS deficiency (Reinecke *et al*, 2009; Singh *et al*, 2020). In fact, direct analysis of 14 HKC strains of Black versus White individuals for mitochondrial electron transfer chain (ETC) activity showed consistently higher levels in cells of the latter group, accompanied by higher ATP production and mitochondrial ROS levels (Fig 8A). In addition, HKCs from White individuals displayed a higher rate of fatty acid β-oxidation, a key OXPHOS-fueling pathway (Rodriguez-Enriquez *et al*, 2015) (Fig 8A). These differences among HKC strains were maintained upon passaging (Appendix Fig S6A), indicating a stable metabolic variance between HKCs of the two ancestry groups. The OXPHOS chain involves the sequential and coordinate electron transfer by five mitochondrial respiration complexes. Further analysis showed that the activity of complex II and III was selectively

higher in the White versus Black HKCs, while activity of the other complexes was similar between the two groups of HKCs (Appendix Fig S6B).

Similar measurements of mitochondrial activity as the ones employed above were performed with several HKC strains as well as SCC13 cells plus/minus *HSD17B7* overexpression. As shown in Fig 8B and C, ETC activity, ATP, and mitochondrial ROS production were all suppressed as a consequence of increased HSD17B7 levels, reproducing the differences observed between HKCs from Black versus White HKCs. Converse effects were caused by *HSD17B7* gene silencing in SCC cells, which resulted in a parallel increase of ETC, ATP, and mtROS production (Fig 8D).

Thus, there are consistent differences in mitochondria OXPHOS activity in HKCs from individuals of Black versus White ancestries, which are inversely related to OXPHOS gene expression and paralleled by modulation of HSD17B7 levels.

### Mitochondrial OXPHOS is under control of HSD17B7 activity and zymosterol production

To assess whether the impact of HSD17B7 on proliferative potential as well as mitochondrial OXPHOS is dependent on its enzymatic activity, we constructed an *HSD17B7* mutant cDNA with two single amino acid substitutions predicted to disrupt its substrate(s) binding sites (Fig 9A and Appendix Fig S7). Unlike intact HSD17B7, overexpression of this mutant had no effect on SCC clonogenicity (Fig 9B), with a similar lack of effects on mitochondrial ETC and ATP and ROS production (Fig 9C and Appendix Fig S5D).

A distinguishing feature of the HSD17B7 enzyme, not shared by other family members that may compensate for its other functions, is that it catalyzes production of zymosterol, a key element in the cholesterol biosynthesis chain (Saloniemi *et al*, 2012). To assess whether this enzymatic product is involved in control of the OXPHOS reaction, SCC cells plus/minus *HSD17B7* gene silencing were treated with zymosterol versus ethanol vehicle control. While *HSD17B7* knockdown resulted in the expected increase of ETC, ATP, and mitochondrial ROS production, all these parameters were brought to control levels by zymosterol treatment (Fig 9D and Dataset EV3 for all metabolic measurements of Figs 8 and 9).

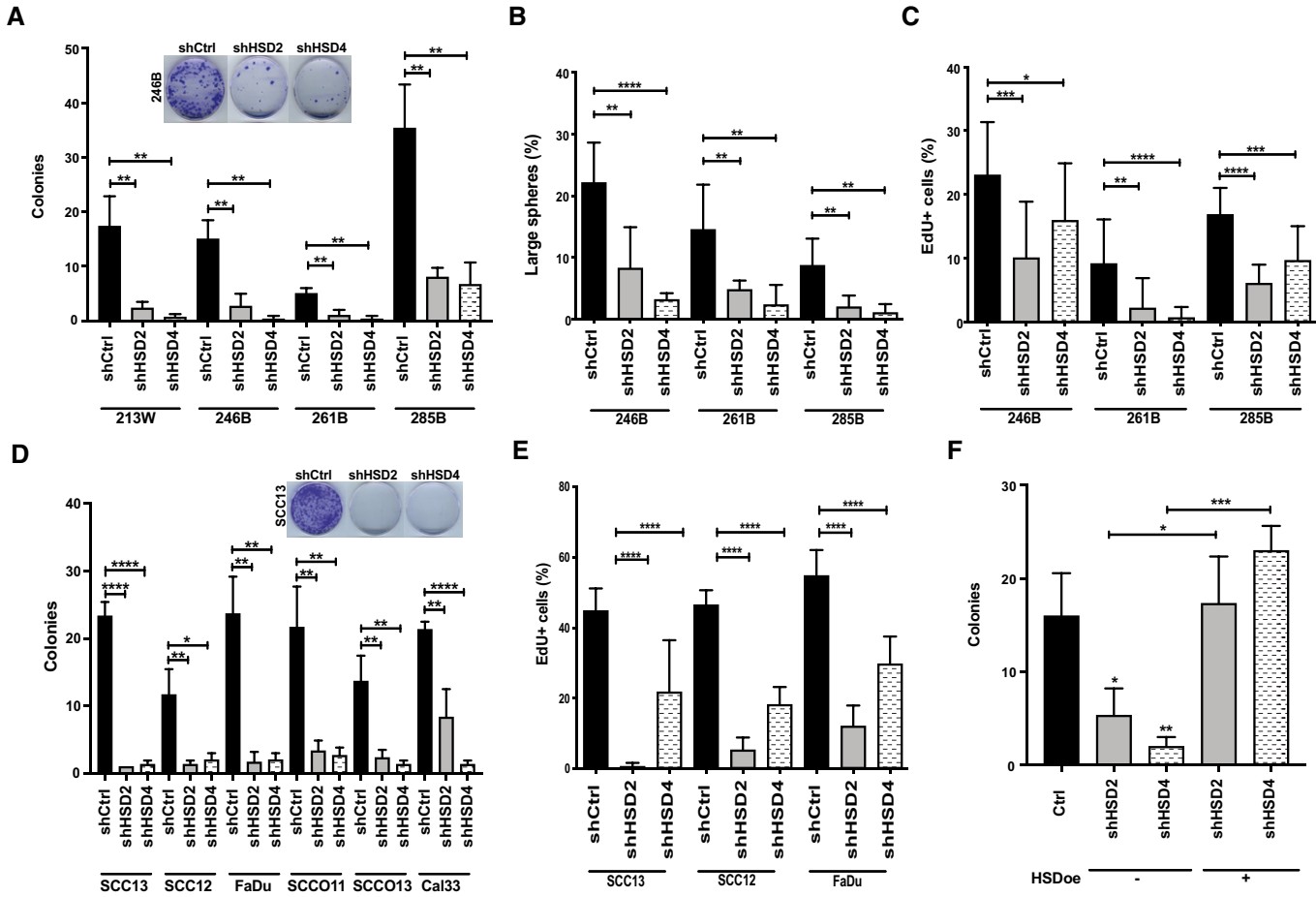

**Figure 6. *HSD17B7* plays an essential function in HKC and SCC cell proliferation.**

A   Colony-forming assays of multiple HKC strains from donors of White (213W) and Black (246B, 261B, 285B) origin, which were infected with two *HSD17B7*-silencing lentiviruses (shHSD2, shHSD4) versus vector control (shCtrl). Gene silencing efficiency is shown in Appendix Fig S6. Cells were plated in triplicate dishes at limited density (1,000 cells per 60-mm dish) at 5 days after infection and fixed a week later for determination of colony formation as in Fig 5A. Shown are representative dishes together with data quantification, expressed as numbers of colonies ± SD. *n*(dishes per condition) = 3. **P < 0.01. 2-tailed unpaired *t*-test.

B   Sphere-forming assays of multiple HKC strains plus/minus *HSD17B7* silencing as in the previous panel. Cells were cultured in Matrigel in triplicate dishes, utilizing the same conditions as for the experiments of Fig 2F. Data are expressed as ratio of large spheres (> 2,000 pixels ≥ 0.0095 mm$^2$) versus total number of spheres (> 100 pixels ≥ 0.00047 mm$^2$). ± SD. *n*(dishes per condition) = 3. **P < 0.01; ****P < 0.0001, 2-tailed unpaired *t*-test.

C   EdU labeling assays of multiple HKC strains plus/minus *HSD17B7* silencing as in the previous panels. Cells were plated in triplicate dishes 5 days after lentiviral vector infection and selection and pulse labeled with EdU for 3 h. Shown are % of EdU-positive cells ± SD. *n*(dishes per condition) = 3. *P < 0.05, **P < 0.01; ***P < 0.0005; ****P < 0.0001, 2-tailed unpaired *t*-test.

D   Colony-forming assays of the indicated SCC cell lines plus/minus *HSD17B7* silencing as in the previous panels. Cells plated in triplicate dishes 5 days after lentiviral vector infection and selection were tested for clonogenic capability as in (A). Data are represented as number of colonies ± SD. *n*(dishes per condition) = 3. *P < 0.05, **P < 0.01; ****P < 0.0001, 2-tailed unpaired *t*-test.

E   EdU labeling assays of the indicated SCC cell lines plus/minus *HSD17B7* silencing. Experimental conditions were as in (C). *n*(dishes per condition) = 3. ****P < 0.0001, 2-tailed unpaired *t*-test. Data are represented as % of EdU-positive cells ± SD.

F   Colony-forming assays of SCC13 cells plus/minus lentivirally mediated *HSD17B7* overexpression and subsequent infection with two shRNA silencing vectors as indicated. Experimental conditions were as in (D). *n*(dishes per condition) = 3. *P < 0.05, **P < 0.01; ***P < 0.0005, 2-tailed unpaired *t*-test. Data are represented as number of colonies ± SD.

---

Thus, mitochondrial OXPHOS levels are under control of *HSD17B7* activity, with zymosterol as likely effector.

## Discussion

A much greater genetic variability is known to exist among individuals within any given ancestry than among ancestries. Yet, large-scale genotyping efforts have revealed differences of functional significance among populations of various origin at the level of restricted loci. For instance, analysis of the HapMap project, based on genotype analysis from Caucasian, African, and Asian population samples, has pointed to a small set of SNPs of predictive value for skin and eye color differences among populations (Huang *et al*, 2015). Our combined evidence, stemming from analysis of keratinocytes and keratinocyte-derived tumors from individuals of Black

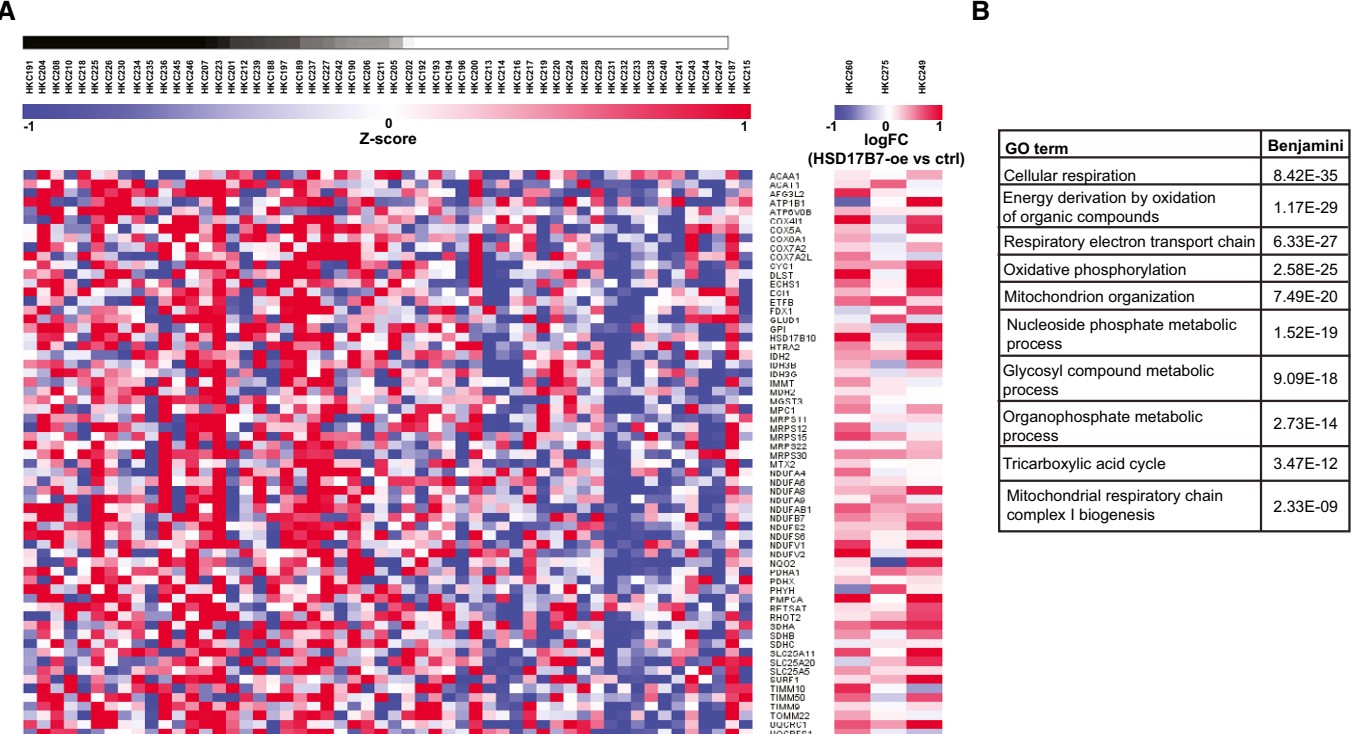

**Figure 7. Expression of mitochondrial function-related genes is higher in Black than White HKCs and upregulated by HSD17B7 overexpression.**

A   Top bar: gradient of population admixture of the indicated HKC strains of Black versus White individuals, on the basis of the SNP genotype analysis shown in Fig 1B, with the simplest assumption of two ancestries ($K = 2$). Gradient ranges from black to white colors for Black and White individuals with no population admixture, respectively. Individuals with increasing proportions of White population admixture are represented with degrading shades of gray. Left: heatmap of expression levels of the indicated genes from the OXPHOS gene signature of Fig 3C, in Black versus White HKC strains ordered according to genome admixture calculations and in three strains of White HKCs with *HSD17B7* overexpression. Modified Z scores of the individual genes, as median-centered log2 intensity values divided by standard deviation, are shown by color gradient variations. Right: heatmap of expression levels of the same genes in three White HKC strains infected with an HSD17B7 overexpressing lentivirus versus empty vector control as in Fig 5A (strains: 260, 275, 249). Expression levels are derived from global transcriptomic analysis (Clariom D hybridization; GEO accession number GSE172288) and expressed as logFold change values of treated versus control samples.

B   Gene Ontology (GO) analysis of genes analyzed in the previous panel. Shown is a list of process networks with statistical significance of enrichment in Black HKCs. The complete list is provided in Dataset EV1.

African versus Caucasian ancestries, has led to a differentially expressed gene of unsuspected importance in control of keratinocyte stem cell and oncogenic potential as well as mitochondrial OXPHOS activity: the *HSD17B7* gene, coding for a targetable enzyme at the intersection between sex steroid and cholesterol biosynthesis (Ferrante *et al*, 2020). This gene is a likely co-determinant of the observed differences between keratinocytes of the two ancestries in concert with other as yet to be determined factors.

*Homo Sapiens* originated in Africa about 200–300,000 years ago and appears to have populated other continents > 100,000 years

**Figure 8. Differential mitochondrial OXPHOS activity in Black versus White HKCs and as a function of *HSD17B7* expression.**

A   Purified mitochondrial preparation from multiple HKC strains from White and Black donors (12 per ancestry; passage 2,3; triplicate cultures) were analyzed for levels of electron transport chain activity (ETC, nmol of reduced cytochrome c/min/mg protein), ATP (nmol/mg protein), and ROS (MtROS nmol/mg mitochondrial protein). In parallel, acid-soluble metabolites (ASM, pmol/h/mg protein) levels were measured as a readout of β-oxidation. Similar measurements of HKCs strains at a later passage are shown in Appendix Fig S6. Data are displayed as average values for all tested HKC strains (3 cultures per strain for ETC, ATP, and FAO, 1 culture per strain for mtROS, white and black dots, depending on ancestry) together with mean ± SD. n(HKC strains per ancestry) = 12. ***$P$ < 0.0005; ****$P$ < 0.0001, 2-tailed unpaired *t*-test. Individual experimental values are provided in Dataset EV3.

B   Multiple HKC strains (249W, 260W, 275W) stably transduced with an *HSD17B7*-expressing lentivirus (HSD-oe) versus empty vector control (Ctrl) were analyzed as in (A). Data are shown as individual values for 3 parallel cultures per strain together with mean ± SD. *$P$ < 0.05; **$P$ < 0.01; ***$P$ < 0.0005; 2-tailed unpaired *t*-test. Individual experimental values are provided in Dataset EV3.

C, D   SCC13 cells stably infected with an *HSD17B7*-expressing lentivirus (HSD-oe) versus empty vector control (Ctrl) (C) or two *HSD17B7*-silencing lentivirus (shHSD2, shHSD4) versus vector control (shCtrl) (D) were analyzed as in the previous panels. Shown are individual values for 3 parallel cultures per condition together with mean ± SD. **$P$ < 0.005; ***$P$ < 0.0005; ****$P$ < 0.0001. 2-tailed unpaired *t*-test. Additional metabolic measurement experiments of SCC cells plus/minus *HSD17B7* overexpression and silencing, in the context of additional manipulations, are shown in Fig 9.

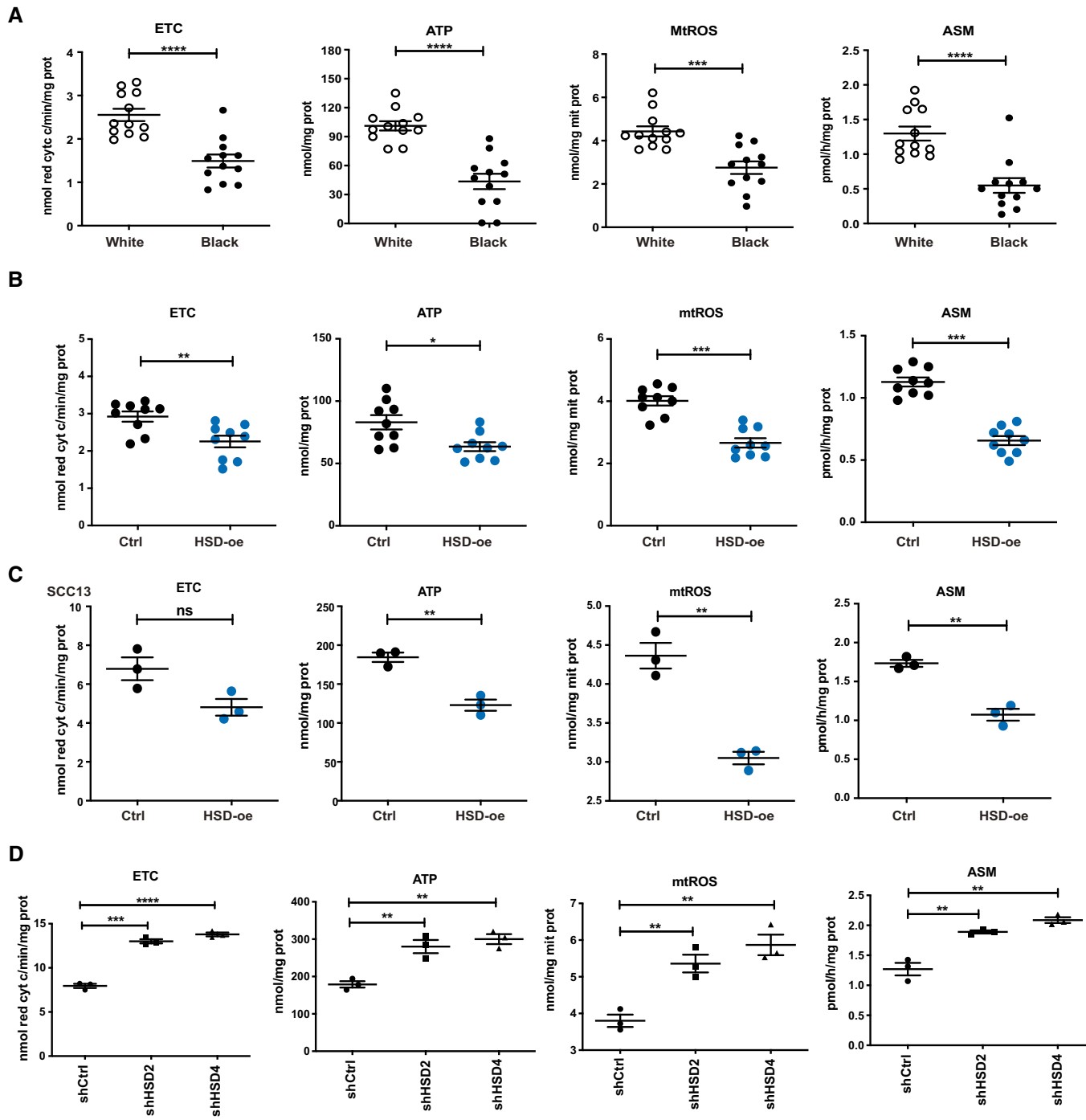

**Figure 8.**

later, through two "bottle neck" migrations of relatively small subpopulations (Skoglund *et al*, 2017; Mathieson, 2020). This "out of Africa" hypothesis (Ashraf & Galor, 2013) can help explaining the greater genetic variation of African populations versus those who expanded in the Eurasian, Australian, and American continents (Choudhury *et al*, 2020). In spite of this, populations of Black African ancestry have generally a higher susceptibility to many cancer types, suggesting the persistence of commonly linked genetic regions affecting organismic functions as well as cancer cells of

origin and their microenvironment (Ozdemir & Dotto, 2017). A large-scale genomic analysis across tumor types has revealed significant ancestry- and tissue-specific differences of likely functional significance, such as higher *FBXW7* mutation rates in patients of African descent (Carrot-Zhang *et al*, 2020). A recent study of a large panel of cancer cell lines has also yielded ancestry-specific genetic and transcriptomic differences (Kessler *et al*, 2019), and a spectrum of ancestry-specific germline variations with a possible involvement in cancer predisposition has also been reported (Oak *et al*, 2020). By

contrast, integrated genomic and functional analysis of normal cells from which cancers originate is, to our knowledge, still missing.

By focusing on keratinocytes, from which SCCs arise, we have found that cells derived from individuals of Black African versus Caucasian ancestries are generally endowed with intrinsically higher oncogenic and self-renewal potential. The use of foreskin-derived HKCs from a cohort of young boys of Black African versus Caucasian origin eliminated confounding variables such as sex and age. GSEA of the transcriptomic profiles of HKC strains of the two ancestries showed that, out of an established hallmark of 50 gene signatures, there was a significant association with only one related to the OXPHOS process. Direct analysis of mitochondrial electron transfer chain and ATP and ROS production showed that they were

all consistently higher in HKC strains from White versus Black donors, which is of likely functional significance given the positive link between increased mitochondrial respiration and ROS production and exhaustion of stem cell potential and differentiation in a variety of cell types (Ito *et al*, 2006; Tothova *et al*, 2007; Owusu-Ansah & Banerjee, 2009; Tormos *et al*, 2011), including keratinocytes (Hamanaka *et al*, 2013; Bhaduri *et al*, 2015).

Integrated transcriptomic analysis of HKC strains and keratinocyte-derived HNSCCs from Black versus White individuals was used to identify differentially expressed genes of functional significance. A number of genes were found, mostly involved in metabolic and detoxifying functions, which will have to be further pursued. Here, we focused on *HSD17B7*, the top-ranked gene

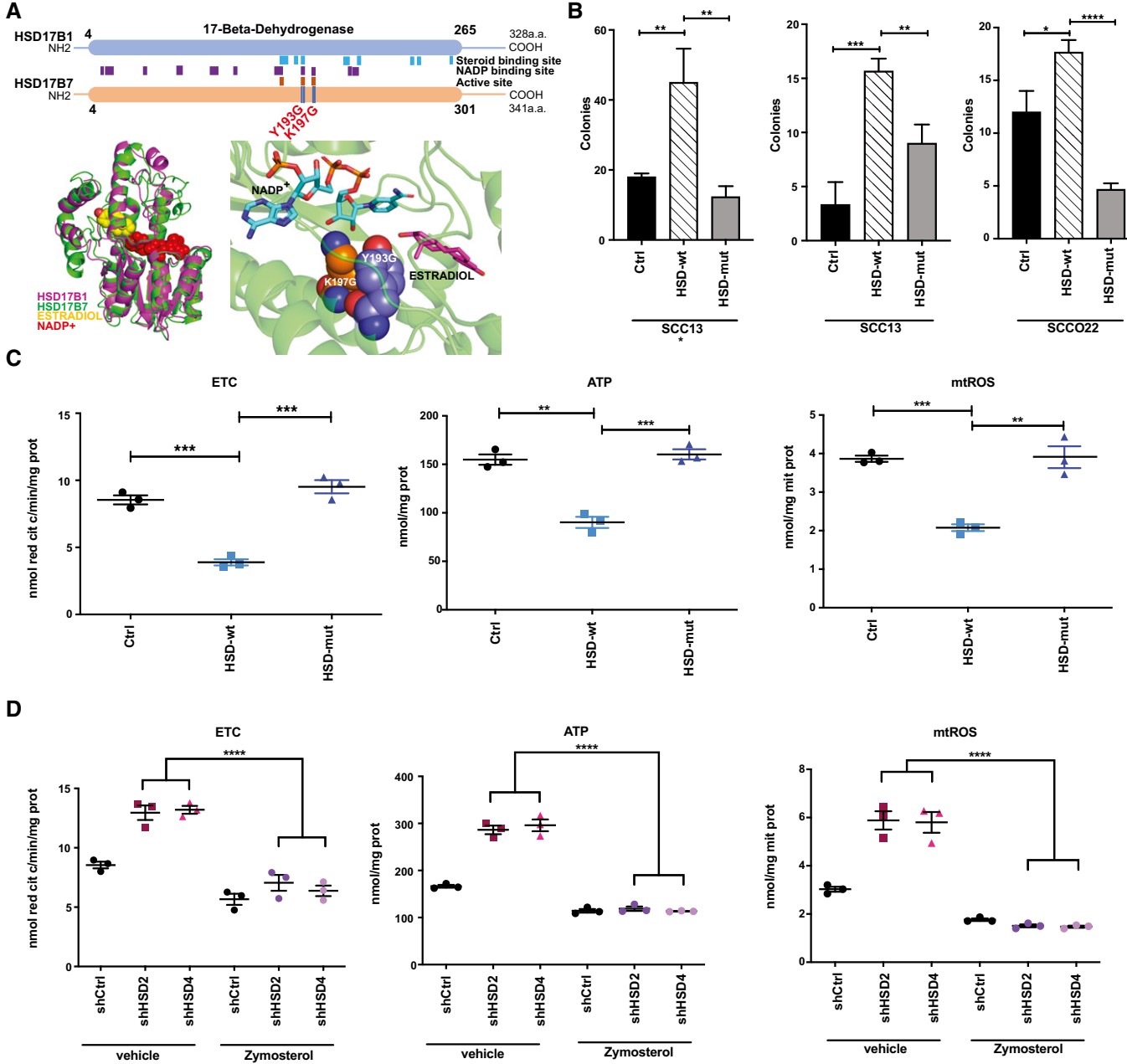

Figure 9.

**Figure 9. HSD17B7 functions are dependent on catalytic activity and involve zymosterol production.**

A    Upper panel: schematic representation of the 17-beta-dehydrogenase domain shared between the HSD17B1 and HSD17B7 enzymes with positions of the NADP and steroid-binding sites and the active site (Breton *et al*, 1996). Two conserved charged residues in these domains, Tyrosine (Y) 193 and Lysine (K) 197, were changed into Glycine (G). Lower panel: close-up views of active site of HSD17B7 showing the NADP and steroid-binding pocket and position of the Y193G and K197G amino acid substitutions.

B    Clonogenicity assays with two different SCC cell lines stably infected with lentiviruses overexpressing *HSD17B7* in wild type (HSD-wt) or Y193G and K197G mutated (HSD-mut) forms versus empty vector control (Ctrl). SCC13 cells were cultured in either DMEM + 10% FBS (*) or HKC growth media. *HSD17B7* overexpression was verified by RT-qPCR and immunoblot analysis shown in Appendix Fig S6. Experimental conditions are as in Fig 5A. Data are represented as number of colonies (> 0.149mm²) ± SD. *n*(dishes per condition) = 3. *$P < 0.05$, **$P < 0.005$; ***$P < 0.0005$, ****$P < 0.0001$. 2-tailed unpaired *t*-test.

C    Purified mitochondrial preparation from SCC13 cells infected with lentiviruses expressing wild-type (HSD-wt) and Y193G and K197G mutated (HSD-mut) forms versus empty vector control (Ctrl) as in (B) were assayed for levels of electron transport chain activity, ATP and mtROS production as in Fig 8A. Data are displayed as average values of triplicate measurements together with mean ± SD. *n*(dishes per condition) = 3. **$P < 0.005$; ***$P < 0.0005$. 2-tailed unpaired *t*-test. Individual experimental values are provided in Dataset EV3.

D    SCC13 cells were infected with two *HSD17B7*-silencing lentiviruses (shHSD2, shHSD4) versus vector control (shCtrl) followed, 3 days after infection and antibiotic selection, by treatment with zymosterol (2.5 μM) or ethanol vehicle alone for 48 h. Purified mitochondrial preparation was assayed for levels of electron transport chain activity, ATP, and mtROS production as in Fig 8A. Data are displayed as average values of triplicate measurements together with mean ± SD. *n*(dishes per condition) = 3. ****$P < 0.0001$, 2-tailed unpaired *t*-test. Individual experimental values are provided in Dataset EV3.

according to statistical significance in both HKC and HNSCC data sets, with higher expression associated with poor HNSCC patients' survival across ancestries. A number of eQTLs linked with differential *HSD17B7* expression are present in the Black African versus Caucasian populations. One of these maps within a binding site of pro-oncogenic transcription factors of the ETS family (Sizemore *et al*, 2017), with the Black-specific SNP allele maintaining the recognition sequence and the White-specific SNP disrupting it (Fig 4H). Consistent with an involvement of ETS transcription factors in control of *HSD17B7* expression, this gene is part of transcriptional network described in ETV1-overexpressing prostate cancer cells (Baena *et al*, 2013). In the present context, ETS transcription factors are likely to function in concert with other determinants of gene expression, as the cognate eQTL co-segregates with other ancestry-specific eQTLs, including one affecting the recognition sequence for transcription factors of the Fox family (Fig 4H), which have also been reported to play a role in cancer development (Tong & Hu, 2021).

HSD17B7 belongs to the family of hydroxysteroid (17β) dehydrogenases (HSD17Bs), which catalyze conversion of lowly active ketonated steroid sex hormones into highly active hydroxylated forms (Saloniemi *et al*, 2012). In addition to this shared function, HSD17B7 has been selectively reported to function as a 3-ketosteroid reductase (3KSR) essential for cholesterol biosynthesis (Breitling *et al*, 2001). An essential role of HSD17B7, which cannot be complemented by other HSD17B family members, was demonstrated by its loss-of-function impact in HKCs and SCC cell proliferative potential as well as OXPHOS activity. Converse effects resulting from HSD17B7 overexpression reproduced the differences of cells from individuals of Black versus White ancestries. Expression of an HSD17B7 mutant with disruption of its substrate-binding sites failed to affect proliferative potential of cells and OXPHOS levels, indicating that the HSD17B7 enzymatic activity is most likely involved. In fact, the increased ETC activity and ATP and ROS levels in cells with *HSD17B7* gene silencing were brought back to control levels by treatment with zymosterol, a specific product of the HSD17B7 enzymatic reaction not shared with other family members. While lower than in other subcellular compartments, cholesterol levels are important determinants of the biophysical and functional properties of mitochondrial membranes, negatively controlling assembly of

respiratory super-complexes and OXPHOS activity (Budin *et al*, 2018; Solsona-Vilarrasa *et al*, 2019). While HSD17B7-dependent zymosterol and cholesterol production can therefore have a direct impact on mitochondria, they could also function as part of a bidirectional mitocellular communication network (Mottis *et al*, 2019), which has also been implicated in cancer disparities among populations with different ancestries (Choudhury & Singh, 2017).

Overall our findings point to the importance of integrated genetic and functional studies for probing into different cancer susceptibility of various human populations, yielding insights and specific targets for more personalized approaches to cancer prevention and treatment from which all individuals may benefit across ancestries.

# Materials and Methods

### Patients' samples and primary cells

Discarded human foreskin samples were obtained from the Pediatric Surgery Department of the Lausanne University Hospital (department de chirurgie pédiatrique of the Centre Hospitalier Universitaire Vaudois) with Human Ethics institutional review board approval and signed formularies of informed patients' consent and self-reported country of origin (UNIL; protocol # 222-12). Informed consent was obtained from all subjects (or from their parents if minors), and all the experiments were conformed to the principles set out in the WMA Declaration of Helsinki and the Department of Health and Human Services Belmont Report.

After rinsing in Hank's balanced salt solution (HBSS, Gibco, Thermo Fisher Scientific) and removal of connective tissue, fat, and mucosa, foreskin samples were digested overnight at 4°C in 10 mg/ml Dispase II (Roche) solution in HBSS. After physical separation from dermis, epidermis was chopped into small pieces and digested by 0.25% Trypsin–EDTA (Gibco) for 1 hr at 4°C. After adding equal volume of serum-free keratinocyte medium (Gibco, Thermo Fisher Scientific) supplemented with EGF (0.2 ng/ml), 1% antibiotics, and 25 μg/ml bovine pituitary extract (BPE), named "complete SFM", cell suspensions were filtered through a sterile gaze and cell pellets were collected by centrifugation. Re-suspended cells were plated at a concentration of $1 \times 10^5$ cells/10-cm dish in complete SFM

medium. Cultured cells were split when confluency reached 50% (around 4 days after plating), and the same regimen was used for further amplification.

## SCC cell lines

Skin and oral SCC cells were obtained and cultured as previously described (Al Labban et al, 2018). Oral SCC cells (SCCO11, SCCO13, and SCCO22) were provided by James Rocco (Massachusetts General Hospital, Boston, Massachusetts, USA). H/NSCC cells (Cal33 and FaDu) were provided by Genrich Tolstonog (CHUV, Lausanne, CH). Skin SCC cells (SCC12 and SCC13) were provided by James Rheinwald (Brigham and Women's Hospital, Boston, Massachusetts, USA). SCC cells used for experiments in Fig 5B and SCC13* in Fig 9B were cultured and tested in DMEM supplemented with 10% fetal bovine serum and 1% antibiotics, while in all other cases, SCC cells were cultured and tested in SFM. All lines were tested monthly for mycoplasma contamination (by PCR).

## Animal models

All mice were housed, bred, and subjected to listed procedures in the animal facility of the University of Lausanne with institutional board approval. Female NOD/SCID mice (7–10 weeks) were used for xenograft experiments. Mice were obtained from the internal Mouse Facility of the University of Lausanne.

## Cell manipulations and plasmid construction

Conditions for cell culturing and viral infections were as previously reported (Al Labban et al, 2018). Briefly, HEK293 cells were co-transfected with viral packaging and lenti- or retro-viral vector plasmids using Jetpei (polyplus transfection) transfection reagent for 6 h, with virus collection 48 h later. HKCs and SCCs were infected for 1 and 2 h, respectively, with high-titer lenti- and retroviruses in DMEM complete medium supplemented with 8 μg/ml polybrene (Sigma-Aldrich) for 1 h, with further culturing in complete SFM in the case of HKCs and SCC cells as indicated. 48 h after infection, cells were selected for puromycin resistance (1 μg/ml) for 3 days, prior to further testing.

The HSD17B7 overexpressing lentiviral vector (HSD-oe) was made by cloning into pLV-CM-puro vector the HSD17B7 open reading frame obtained by PCR amplification of another expression vector purchased from the EPFL gene expression core facility – www.epfl.ch/research/facilities/gene-expression-core-facility/- NCBI Gene ID: 51478, ORF in pLX304 vector with V5 at C-term of the protein). The Infusion KIT (Takara Bio) was used for cloning. The lentiviral vector expressing mutant HSD17B7 (HSD-mut) was obtained from the HSD17B7-wt sequence inserted into pLV-CM-puro, by 2 single point mutations resulting in the Y193G, K197G amino acid substitutions. The QuickChange Site-Directed Mutagenesis Kit (Agilent) was used with the forward primer : 5'-CAGCAAA GGCAAGGAACCCGGCAGCTCTTCCGGATATGCCACTGACCTTTTG-3', and the reverse primer : 5'-CAAAAGGTCAGTGGCATATCCGGAA GAGCTGCCGGGTTCCTTGCCTTTGCTG-3'.

Lentiviral vectors encoding shRNA against HSD17B7 were purchased from Sigma (MISSION shRNA, lentiviral backbone PLKO, cat # TRCN0000027146 for shHSD2, TRCN0000027138 for shHSD4).

## Cell assays

For clonogenicity assays, cells were plated in triplicate onto 60-mm dishes at low density (1,000 cells/dish) and cultured for 8–10 days. Colonies were fixed with 4% formaldehyde and stained with 0.1% crystal violet. The number and size of colonies were counted by Fiji software.

For sphere-forming assays, cells were plated onto 8-well chamber slides (1,000 cells/well) pre-coated with 80 μl of Matrigel (BD Biosciences) for 1 h at 37°C. After 8 days, slides were fixed with 4% formaldehyde for 20 min at RT and images were acquired using EVOS inverted microscope (Thermo Fisher Scientific). Number and size of spheres were counted by Fiji software.

For live-cell proliferation assays, cells were plated in triplicate at low density (1,000 cells/well) into 96-well trays. Plates were placed into IncuCyte ZOOM (Sartorius) for 5 days, with pictures and cell density measurements taken every 2–3 h.

5-Ethynyl-2′-deoxyuridine (EdU) incorporation assays were performed using Click-iT Plus EdU Imaging Kit (Thermo Fisher Scientific) following the manufacturer's instructions, with nuclear staining by DAPI (Sigma, D9542, 1:1,000 dilution in PBS, 10 min incubation). Fraction of EdU-positive cells was determined by Fiji software.

## Intradermal tumorigenicity assays

Assays were carried out in 7- to 10-week-old NOD/SCID mice (females). Early passage HKCs derived from Black or White donors, or HKCs plus/minus HSD17B7 overexpression, were infected with a retrovirus expressing the dominant-negative p53$^{R248W}$ protein (pBABE-puro-p53$^{R248W}$; (Okawa et al, 2007)) and selected for puromycin resistance (1 μg/ml) for 4 days. They were then superinfected with an oncogenic H-ras$^{V12}$ expressing retrovirus (LZRS-H-ras$^{V12}$; (Dajee et al, 2003)) for 24 h prior to in vivo testing.

Genetically manipulated HKCs and SCC13 cells were brought into suspension, admixed with Matrigel (BD Biosciences, 70 μl), and injected in equal numbers ($10^5$ cells per injection) into left and right sides of mouse back. Mice were sacrificed 15 days after injection for tissue analysis. Tumor size was measured using digital caliper, and volume was calculated using the following formula: volume = length × width × height.

## Immunodetection

Conditions for immunofluorescence, immunohistochemistry, and immunoblotting were as previously reported (Al Labban et al, 2018). For immunofluorescence, cells were fixed in 4% paraformaldehyde and permeabilized in 0.1% Triton X-100, followed by blocking in 5% donkey serum in PBS. Primary antibodies incubation was at 4°C overnight (SLC25A5, Atlas Antibodies cat. no. HPA046835, 1:20 dilution), followed by secondary antibodies incubation at RT for 1 h, nuclear DAPI staining, and mounting in Fluoromount Mounting Medium (Dako). Mitotracker Red CMXRos staining was performed following manufacturer's instruction and using a working concentration of 250 nM.

Frozen tumor sections (7–8 μm) were fixed in 4% paraformaldehyde and permeabilized in 0.2% Triton X-100, followed by blocking in 5% BSA in PBS. Primary antibodies for MKI67 (GeneTex, GTX20833, 1:500 dilution), K1 and K10 (Covance, PRB-149P and PRB-

159P, 1:500, and 1:800 dilution, respectively), and pan-keratin (BMA Biomedicals AG, T-1302, 1:500 dilution) were incubated at 4°C overnight, followed by incubation with secondary antibodies conjugated with Alexa Fluor fluorescent dyes (Alexa Fluor 568 or Alexa Fluor 488, 1:300 and 1:800 dilution, respectively) (Thermo Fisher Scientific) at RT for 1 h and nuclear DAPI staining. Sections were mounted in Fluoromount Mounting Medium (Dako, 1:1,000 dilution).

Immunofluorescence images were acquired with a Zeiss Axiovision or Zeiss LSM880 confocal microscope. Axiovision or Zen Black software was used for acquisition and processing of images. For fluorescence signal quantification, acquired images for each color channel were imported into ImageJ and quantified using the functions "measurement" or "particle analysis" for selection of areas or cells of interest.

For immunoblotting, cells were lysed in RIPA lysis buffer (10 mM Tris–Cl, pH 8.0, 1 mM EDTA, 1% Triton X-100, 0.1% sodium deoxycholate, 0.1% SDS, 140 mM NaCl, supplemented with 1% phosphatase inhibitor cocktail 2—Sigma, P5726—and Complete Mini EDTA-free protease inhibitor—Roche, 11836170001). Equal protein amounts—determined by quantification with BCA assay (Pierce BCA protein assay kit—Thermo Fisher Scientific #23225)—were subjected to 10% SDS–PAGE followed by immunoblot analysis, with sequential probing with antibodies against HSD17B7 (Santa Cruz, sc-393936, diluted 1:500 in TBS 5%BSA solution) and vinculin (HRP conjugate; Cell Signaling Technology, #18799, diluted 1:2,000 in TBS 5% BSA solution) and corresponding secondary antibody (anti-mouse HRP diluted 1:2,000 in TBS 5% milk solution), with Super Signal West Pico PLUS Chemiluminescent Substrate (Thermo Fisher Scientific, cat. # 34580) for detection.

### RT-qPCR analysis

For RT-qPCR analysis, total RNA samples (1ug) were reverse-transcribed into cDNA using RevertAid H Minus Reverse Transcriptase (Thermo Fisher Scientific). Real-time qPCR was by SYBR Fast qPCR Master Mix (Kapa Biosystems) and a Light Cycler 480 (Roche). All RNA samples were analyzed in triplicate with gene-specific primers and 36B4 for normalization. The primers list is the following:

HSD17B7—FWD: 5'-TCTAAATGCTGGGATCATGCC-3'
HSD17B7—REV: 5'-AACACCTCCTGAAGTCCATCA-3'
36B4—FWD: 5'-GCAATGTTGCCAGTGTCTGT-3'
36B4—REV: 5'-GCCTTGACCTTTTCAGCAAG-3'

### Transcriptomic analysis

Total RNA extraction and purification were using the GeneChip® WT PLUS Reagent Kit (Thermo Fisher Scientific). RNA samples (500 ng, OD260/OD280 ≥ 1.8, RIN ≥ 8) were processed for probe preparation and hybridization to human Clariom™ D Arrays (Thermo Fisher Scientific) carried out at the Institute of Genetics and Genomics of Geneva (iGE3). Data processing and analysis were utilizing the Transcriptome Analysis Console (TAC) Software (Thermo Fisher Scientific).

PANTHER Hidden Markov Models online software was used for protein classification. DAVID software was used for gene ontology analysis. Gene Set Enrichment Analysis (GSEA) of transcriptomic profiles was by GSEA v4.1.0, with curated gene sets were obtained from the MSigDB v7.2 database (https://software.broadinstitute.org/cancer/software/gsea). *HSD17B7* gene expression correlation analysis was carried out with the R *stats* package (*cor* function, Spearman's correlation, threshold = 0.6). Raw and processed transcriptomic data were deposited on the GEO Omnibus database (GSE156011 for Black and White samples, GSE172288 for White HKCs with HSD17B7 overexpression).

### Genotyping and SNP analysis

Genomic DNA samples from 62 foreskin fibroblasts, prepared in parallel with the HKC strains, were analyzed by SNP array hybridization utilizing a HumanOmni2.5-8 Beadchip (www.illumina.com). PLINK 1.07 and GCTA analysis tools were used for data quality control and processing. Two samples with low genotyping rate (< 95%) were removed. After data quality control (call rate ≥ 0.95, minor allele frequency (MAF)> 0.05 and LD threshold = 0.2), 53837 SNPs were used for PCA. The package "snpStats" was used as statistical tool for SNP association studies, and the package 'SNPRelate' (Zheng *et al*, 2012) was used to calculate the eigenvectors and eigenvalues for principal component analysis (PCA) based on filtered SNPs. The first two principal components were used to create the PCA plot. Raw and processed genotyping data were deposited on the GEO Omnibus database (GSE156977).

### EQTLs and mapping SNPs to transcription factor binding sites

Expression quantitative trait loci (eQTL) analysis was done by crossing the genome-wide SNPs data and the HKC transcriptomic profiles, annotated from the Genome Reference Consortium Human Build 37 (GRCh37) and 38 (GRCh38), respectively. Genome coordinates between the two assemblies were established using the Lift Genome Annotations (UCSC Genome Browser), and eQTL analysis was carried out using the MatrixEQTL R package (Shabalin, 2012), utilizing matrix operations that include linear regression with additive genotype effects. Local association tests were performed for all variants that lie within an a priori-defined window (1MB) of the *HSD17B7* gene, setting false-positive rate (FDR) correction at 0.005. Significant associations represent variants that may have a direct effect on the transcription rate of nearby genes, likely by altering activity of cis-regulatory elements. We used the SNP2TFBS web interface (https://ccg.epfl.ch/snp2tfbs/) to identify variations that affect putative transcription factor binding and, for each cis-eQTL, we examined the fixation index (FST) value between White (EUR) and Black (AFR) populations in the 1,000 genome phase I dataset using the SPSmart online software (Amigo *et al*, 2008). The LDmatrix tool was used to create a heatmap matrix of pairwise linkage disequilibrium statistics of the 6 eQTLs found to be associated with *HSD17B7* expression and with significantly different allele frequency in the Black versus White populations. To calculate linkage disequilibrium, the following reference populations from the 1,000 Genomes Project were used: AFR (African), AMR (Ad Mixed American), EAS (East Asian), EUR (European), SAS (South Asian).

### Mitochondrial extraction

Cells were lysed in 0.5 ml mitochondria lysis buffer (50 mM Tris–HCl, 100 mM KCl, 5 mM MgCl$_2$, 1.8 mM ATP, 1 mM EDTA,

pH 7.2), supplemented with Protease Inhibitor Cocktail III (Sigma), 1 mM phenylmethylsulfonyl fluoride (PMSF), and 250 mM NaF. Samples were clarified by centrifugation at 650 $g$ for 3 min at 4°C. Supernatants were collected and centrifuged at 13,000 $g$ for 5 min at 4°C. The new supernatants, corresponding to the cytosolic fraction, were used for cytosolic ROS measurements. Pellets, containing mitochondria, were washed once with lysis buffer and re-suspended in 0.25 ml mitochondria resuspension buffer (250 mM sucrose, 15 mM $K_2HPO_4$, 2 mM $MgCl_2$, 0.5 mM EDTA). 50 µl aliquots were sonicated and used for the measurement of protein content by the BCA Protein Assay Kit (Sigma) and for quality control, to confirm the presence of mitochondrial proteins in the extracts, by SDS–PAGE and immunoblotting of 10 µg of samples with an anti-porin antibody (Abcam; clone 20B12AF2). The remaining 200 µl of re-suspended mitochondria preparations was used for the metabolic assays reported below.

## Mitochondrial electron transfer chain (ETC) activity

To measure electron flux from complex I to complex III, taken as an index of the mitochondrial respiratory activity (Wibom *et al*, 2002), 50 µg of non-sonicated mitochondrial samples, isolated as indicated above, was re-suspended in 0.2 ml buffer A (5 mM $KH_2PO_4$, 5 mM $MgCl_2$, 5% w/v bovine serum albumin, BSA; pH 7.2) to which 0.1 ml buffer B (25% w/v saponin, 50 mM $KH_2PO_4$, 5 mM $MgCl_2$, 5% w/v BSA, 0.12 mM oxidized cytochrome c, 0.2 mM $NaN_3$, which blocks complex IV allowing the accumulation of reduced cytochrome c; pH 7.5) was added for 5 min at room temperature. The cytochrome c reduction reaction was started by adding 0.15 mM NADH and was followed for 5 min at 37°C, reading the absorbance at 550 nm by a Packard microplate reader EL340 (Bio-Tek Instruments, Winooski, VT). The results were expressed as nanomoles of reduced cytochrome c /min/mg mitochondrial proteins.

More specific spectrophotometric determination of electron flux through complex I, complex II, complex III, and complex IV was measured as detailed in Ref. (Wibom *et al*, 2002). Specifically, to measure complex I activity, 20 µg of non-sonicated mitochondrial samples was re-suspended in 0.2 ml buffer 1A (5 mM $KH_2PO_4$, 5 mM $MgCl_2$, 5% w/v BSA), incubated 1 min at RT followed by 7 min in 0.1 ml buffer 1B (25% w/v saponin, 50 mM $KH_2PO_4$, 5 mM $MgCl_2$, 5% w/v BSA, 0.12 mM oxidized ubiquinone, which acts as electrons shuttle from complex I to complex III, 2.5 mM antimycin A, which inhibits complex III, 0.2 mM $NaN_3$, which blocks complex IV; pH 7.5). 1.5 mM NADH, as electron donor was added to the mix. The rate of NADH oxidation was followed for 5 min at 37°C, reading the absorbance at 340 nm. The results were expressed as nanomoles of $NAD^+$/min/mg mitochondrial proteins.

Complex II activity was measured as rate of electrons transfer between complex II and complex III. 20 µg of non-sonicated mitochondrial samples was re-suspended in 0.1 ml buffer 2A (50 mM $KH_2PO_4$, 7.5 mM $MgCl_2$, 25% w/v saponin, 20 mM succinic acid; pH 7.2) and incubated for 30 min at room temperature. 0.2 ml buffer 2B (50 mM $KH_2PO_4$, 7.5 mM $MgCl_2$, 5% w/v BSA, 30 mM succinic acid as substrate of complex II, 0.12 mM oxidized ubiquinone as electrons shuttle from complex II to complex III, 0.12 mM oxidized cytochrome c as acceptor of electrons flowing from complex II to

complex III, 5 mM rotenone to prevent electron flux from complex I, 0.2 mM $NaN_3$, to block complex IV) was added. The rate of reduction of cytochrome c was measured for 5 min at 37°C, reading the absorbance at 550 nm. The results were expressed as nanomoles of reduced cytochrome c/min/mg mitochondrial proteins.

The activity of complex III was measured in the same samples where the electron flux from complex I to complex III was evaluated. After 1 min from the addition of NADH, as inducer of electrons flow, 5 mM rotenone, which blocks the activity of complex I, was added. The rate of reduction of cytochrome c, which is dependent on the activity of complex III only in the presence of rotenone, was followed for 5 min at 37°C, reading the absorbance at 550 nm. The results were expressed as nanomoles of reduced cytochrome c/min/mg mitochondrial proteins.

To measure the activity of complex IV, the rate of oxidation of cytochrome c (reduced form, generated by complex III) was measured. 20 µg of non-sonicated mitochondrial samples was re-suspended in 0.1 ml buffer 4A (50 mM $KH_2PO_4$, 20 mM succinic acid, 25% w/v saponin; pH 7.2) and incubated 30 min at room temperature. 0.2 ml buffer 4B (50 mM $KH_2PO_4$, 5 mM rotenone, which prevents electron flux from complex I to complex III, 30 mM succinic acid as substrate of complex II and electrons generator, 0.03 mM reduced cytochrome c as acceptor of electrons flowing from complex III to complex IV) was added. The rate of oxidation of cytochrome c was followed for 5 min at 37°C, reading the absorbance at 550 nm. The results were expressed as nanomoles of oxidized cytochrome c/min/mg mitochondrial proteins.

## ATP levels

ATP amounts in mitochondrial extracts were measured with the ATP Bioluminescent Assay Kit (Millipore Sigma), as per the manufacturer's instructions. Results were expressed as nmoles/mg mitochondrial proteins.

## ROS measurement

ROS amounts in mitochondrial and cytosolic extracts were measured using the ROS-sensitive fluorescent probe 5-(and-6)-chloromethyl-2',7'-dichlorodihydro-fluorescein diacetate (CM-$H_2$DCFDA; Sigma) (Gray *et al*, 2013). Cytosolic extracts (100 µg proteins) or mitochondrial extracts (50 µg proteins) were incubated for 30 min at 37°C with 5 µM of CM-$H_2$DCFDA, in 96-well plates. Sample fluorescence was read at 492 nm (λ excitation) and 517 nm (λ emission), using a HT Synergy 96-well microplate reader (Bio-Tek Instruments). Results were expressed as nmoles/mg cytosolic or mitochondrial or proteins.

## Fatty acid β-oxidation

The rate of fatty acid β-oxidation was measured according to (Wang *et al*, 2018), using part of the same cells evaluated in the biochemical assays indicated in the previous paragraphs. Cells were re-suspended in 0.5 ml PBA, and 50 µl aliquots were sonicated and used for protein measurements and normalization. The remaining samples were centrifuged 13,000 $g$ for 5 min at room temperature and re-suspended in 0.5 ml Hepes 20 mM (pH 7.4), containing 0.24 mM fatty acid-free BSA, 0.5 mM L-carnitine, 2 µCi [1-$^{14}$C]

## The paper explained

### Problem

Cancer susceptibility of populations with different genetic backgrounds varies significantly for reasons that cannot be solely attributed to socioeconomical and behavioral factors. Recent studies have unmasked an association of ancestry-specific genetic differences in various cancer types and germline variations in cancer predisposition. However, an integrated genomic and functional analysis of normal cells from which cancers originate was missing.

### Results

By focusing on keratinocytes from which squamous cell carcinomas arise, we have found that cells derived from individuals of Black African are generally endowed with intrinsically higher oncogenic and self-renewal potential than those derived from Caucasian ancestries, which are associated with lower mitochondrial oxidative phosphorylation (OXPHOS) activity and ROS production. Combined transcriptomic analysis of primary keratinocytes (HKCs) and head and neck squamous cell carcinomas (HNSCCs) led to the identification of *HSD17B7* as the most highly expressed gene in samples of Black African versus Caucasian origin, with several ancestry-specific expression quantitative trait loci (eQTLs) as possible explanation. Higher *HSD17B7* expression is overall associated with poor HNSCC patients' survival across ancestries. The gene codes for a member of the hydroxysteroid dehydrogenase family of enzymes involved in sex steroid and cholesterol biosynthesis. By combined gain and loss-of-function experiments, we show that *HSD17B7* plays a positive role in control of keratinocyte stem cell and oncogenic potential while it suppresses OXPHOS activity, with zymosterol, a specific enzymatic product, as likely involved.

### Impact

Our findings point to the importance of integrated genetic and functional studies for probing into different cancer susceptibility among and within various human populations. By this combined approach, we have identified a gene of unsuspected importance in control of keratinocyte stem cell and oncogenic potential coding for a targetable enzyme that could be considered for SCC prevention and treatment across ancestries.

palmitic acid (3.3 mCi/mmol, PerkinElmer, Waltham, MA) and transferred into test tubes that were tightly sealed with rubber caps. In each experimental set, cells were pre-incubated for 30 min with the carnitine palmitoyltransferase inhibitor etomoxir (1 μM) or with the AMP-kinase activator 5-aminoimidazole-4-carboxamide ribonucleotide AICAR (1 mM), as negative and positive controls, respectively. After 2-h incubation at 37°C, 0.3 ml of a 1:1 v/v phenylethylamine/methanol solution was added to each sample using a syringe, followed by 0.3 ml 0.8 N $HClO_4$. Samples were incubated for a further 1 h at room temperature and then centrifuged at 13,000 $g$ for 10 min. Both the supernatants, containing $^{14}CO_2$, and the precipitates, containing $^{14}C$-acid soluble metabolites (ASM), i.e., the main products of fatty acid β-oxidation (Gaster *et al*, 2004), were collected. The radioactivity of each supernatant and precipitate was counted by liquid scintillation. Results were expressed as pmoles/h/mg cell proteins of [$^{14}CO_2$] in the supernatant, used as internal quality control, and as pmoles/h/mg cell proteins of [$^{14}C$-ASM] in the precipitates, as indication of fatty acid β-oxidation. In each experiment, the amount of [$^{14}CO_2$] was always < 10% than the amount of [$^{14}C$-ASM], as recommended (Gaster *et al*, 2004).

### Statistical analysis

Statistical testing was performed using Prism 8 (GraphPad Software). Data are presented as mean ± SD, as indicated in the legends. Statistical significance for comparing two experimental conditions was calculated by two-tailed *t*-tests. For Incucyte experiments, statistical significance was calculated by Pearson R correlation. All *P* values divided by figure are indicated in Appendix Table S1. For tumorigenicity assays, individual animal variability was minimized by contralateral injections in the same animals of control versus experimental combinations of cells. No exclusion criteria were adopted for animal studies and sample collection.

## Data availability

The datasets produced in this study are available in the following databases:

- Genotyping data: Gene Expression Omnibus GSE156977 (https://www.ncbi.nlm.nih.gov/geo/query/acc.cgi?acc = GSE156977)
- Affychip ClariomD data – Black versus White cohort: Gene Expression Omnibus GSE156011 (https://www.ncbi.nlm.nih.gov/geo/query/acc.cgi?acc = GSE156011)
- Affychip ClariomD data – White HKC with HSD17B7 overexpression: Gene Expression Omnibus GSE172288 (https://www.ncbi.nlm.nih.gov/geo/query/acc.cgi?acc = GSE172288)

**Expanded View** for this article is available online.

## Acknowledgements

This study was supported by grants from the European Research Council (26075083), the Swiss National Science Foundation (310030B_176404 "Genomic instability and evolution in cancer stromal cells"), and the NIH (R01AR039190, the content does not necessarily represent the official views of the NIH).

## Author contributions

XX, BT, PO, AK, TP, CR, KL performed the experiments and/or contributed to analysis of the results. PO and GC performed the bioinformatics analysis. J-MJ provided clinical samples. ZK provided help and direction with the genetic analysis. XX, BT, PO, KL, and GPD designed the study, assessed the data, and wrote the manuscript. XX and BT contributed equally to this work.

## Conflict of interest

The authors declare that they have no conflict of interest.

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
