## [Review Process File · EMBO Molecular Medicine]

HSD17B7 gene in self-renewal and oncogenicity of keratinocytes from Black versus White populations

Gian Paolo Dotto, Xiaoying Xu, Beatrice Tassone, Paola Ostano, Atul Katarkar, Tatiana Proust, Jean-Marc Joseph, Chiara Riganti, Giovanna Chiorino, Zoltan Kutalik, and Karine Lefort

DOI: [10.15252/emmm.202114133](https://doi.org/10.15252/emmm.202114133)

Corresponding author: Gian Paolo Dotto (paolo.dotto@unil.ch)

Review Timeline:

Submission Date:	15th Feb 21
Editorial Decision:	19th Mar 21
Revision Received:	29th Apr 21
Editorial Decision:	5th May 21
Revision Received:	19th May 21
Accepted:	20th May 21

Editor: Lise Roth

Transaction Report:

19th Mar 2021

Dear Paolo,

Thank you for the submission of your manuscript to EMBO Molecular Medicine, and please accept my apologies for the delay in getting back to you, which is due to the fact that we were waiting for the report from the third referee. As we have not received this report yet and given that both referees #1 and #2 are overall positive, we prefer to make a decision now in order to avoid further delay in the process. Should referee #3 provide a report shortly, we will send it to you, with the understanding that we would not ask you for further-reaching experiments in addition to the ones required in the enclosed reports.

As you will see from the reports below, the referees acknowledge the interest of the study and are overall supporting publication of your work pending appropriate revisions. Addressing the reviewers' concerns in full will be necessary for further considering the manuscript in our journal. Acceptance of the manuscript will entail a second round of review. EMBO Molecular Medicine encourages a single round of revision only and therefore, acceptance or rejection of the manuscript will depend on the completeness of your responses included in the next, final version of the manuscript. For this reason, and to save you from any frustrations in the end, I would strongly advise against returning an incomplete revision.

When submitting your revised manuscript, please carefully review the instructions that follow below. Failure to include requested items will delay the evaluation of your revision:

- 1) A .docx formatted version of the manuscript text (including legends for main figures, EV figures and tables). Please make sure that the changes are highlighted to be clearly visible.
- 2) Individual production quality figure files as .eps, .tif, .jpg (one file per figure).
- 3) A .docx formatted letter INCLUDING the reviewers' reports and your detailed point-by-point responses to their comments. As part of the EMBO Press transparent editorial process, the point-by-point response is part of the Review Process File (RPF), which will be published alongside your paper.
- 4) A complete author checklist, which you can download from our author guidelines (<https://www.embopress.org/page/journal/17574684/authorguide#submissionofrevisions>). Please insert information in the checklist that is also reflected in the manuscript. The completed author checklist will also be part of the RPF.
- 5) Before submitting your revision, primary datasets produced in this study need to be deposited in an appropriate public database (see <https://www.embopress.org/page/journal/17574684/authorguide#dataavailability>). Please remember to provide a reviewer password if the datasets are not yet public. The accession numbers and database should be listed in a formal "Data Availability" section (placed after Materials & Method). Please note that the Data Availability Section is restricted to new primary data that are part of this study.

6) We would also encourage you to include the source data for figure panels that show essential data. Numerical data should be provided as individual .xls or .csv files (including a tab describing the data). For blots or microscopy, uncropped images should be submitted (using a zip archive if multiple images need to be supplied for one panel). Additional information on source data and instruction on how to label the files are available at .

7) Our journal encourages inclusion of *data citations in the reference list* to directly cite datasets that were re-used and obtained from public databases. Data citations in the article text are distinct from normal bibliographical citations and should directly link to the database records from which the data can be accessed. In the main text, data citations are formatted as follows: "Data ref: Smith et al, 2001" or "Data ref: NCBI Sequence Read Archive PRJNA342805, 2017". In the Reference list, data citations must be labeled with "[DATASET]". A data reference must provide the database name, accession number/identifiers and a resolvable link to the landing page from which the data can be accessed at the end of the reference. Further instructions are available at .

8) We replaced Supplementary Information with Expanded View (EV) Figures and Tables that are collapsible/expandable online. A maximum of 5 EV Figures can be typeset. EV Figures should be cited as 'Figure EV1, Figure EV2" etc... in the text and their respective legends should be included in the main text after the legends of regular figures.

- Additional Tables/Datasets should be labeled and referred to as Table EV1, Dataset EV1, etc. Legends have to be provided in a separate tab in case of .xls files. Alternatively, the legend can be supplied as a separate text file (README) and zipped together with the Table/Dataset file. See detailed instructions here:

9) The paper explained: EMBO Molecular Medicine articles are accompanied by a summary of the articles to emphasize the major findings in the paper and their medical implications for the non-specialist reader. Please provide a draft summary of your article highlighting

10) For more information: There is space at the end of each article to list relevant web links for further consultation by our readers. Could you identify some relevant ones and provide such information as well? Some examples are patient associations, relevant databases, OMIM/proteins/genes links, author's websites, etc...

11) Every published paper now includes a 'Synopsis' to further enhance discoverability. Synopses are displayed on the journal webpage and are freely accessible to all readers. They include a short

stand first (maximum of 300 characters, including space) as well as 2-5 one-sentences bullet points that summarizes the paper. Please write the bullet points to summarize the key NEW findings. They should be designed to be complementary to the abstract - i.e. not repeat the same text. We encourage inclusion of key acronyms and quantitative information (maximum of 30 words / bullet point). Please use the passive voice. Please attach these in a separate file or send them by email, we will incorporate them accordingly.

Please also suggest a striking image or visual abstract to illustrate your article as a png file 550 px-wide x 400-px high.

12) As part of the EMBO Publications transparent editorial process initiative (see our Editorial at <http://embomolmed.embopress.org/content/2/9/329>), EMBO Molecular Medicine will publish online a Review Process File (RPF) to accompany accepted manuscripts.

In the event of acceptance, this file will be published in conjunction with your paper and will include the anonymous referee reports, your point-by-point response and all pertinent correspondence relating to the manuscript. Let us know whether you agree with the publication of the RPF and as here, if you want to remove or not any figures from it prior to publication.

I look forward to receiving your revised manuscript.

Yours sincerely,

Lise Roth

Lise Roth, PhD
Editor
EMBO Molecular Medicine

To submit your manuscript, please follow this link:

Link Not Available

Each figure should be given in a separate file and should have the following resolution:
Graphs 800-1,200 DPI

Photos 400-800 DPI
Colour (only CMYK) 300-400 DPI"

*Additional important information regarding figures and illustrations can be found at <https://bit.ly/EMBOPressFigurePreparationGuideline>

***** Reviewer's comments *****

Referee #1 (Comments on Novelty/Model System for Author):

As described in the review, PCA needs to be moved up to first define genetic ancestry of individuals.

The reviewer does not have a background in experimental oncology and defers that evaluation to others.

Referee #1 (Remarks for Author):

Reviewer comment

Xu & Tassone et al. presented an intriguing study investigating the molecular differences observed in HKCs and HNSCCs or Black African vs. White Caucasian individuals. Combining in vitro, in vivo, and computational analyses, they were able to trace key differences to the HSD17B7 gene that have ancestry-associated eQTL. The manuscript is generally well-written and results well-presented. The reviewer has a background in cancer genomics, but not in experimental biology, and will focus on commenting on those sections.

Major:

1. As the author suggested, they "will be employing the term "ancestry" in reference to individuals with common genetic and phenotypic features rather than "race." The reviewer agree with this approach as race is a social construct, whereas genetic ancestry enables investigation of biological hypotheses. Given that the entire paper focused on comparing HKCs/HNSCCs across African and Caucasian ancestries, the authors should first define the African vs. Caucasian HKCs using genetic PCA (rather than "Black" and "White") before other analyses. This is a common practice in the human genetics field to avoid confounding and inaccuracy of self-reported race/ethnicity or potential samples swaps.

a. Typically admixed individuals (ex. >20% of other ancestry) would be separately analyzed. It seems from Fig 2B that roughly half of the individuals the authors defined as "Black" would be admixed. After first defining the ancestry, the author could either keep the two groups but define the African-ancestry group as "individuals with >x% African ancestry", or more appropriately, separate the African ancestry, admixed, and Caucasian individuals.

2. Why did the author choose HNSC for the human cohort analyses? Are there datasets of other keratinocyte cancers (especially those affecting the skin?)

3. Figure 3C, the survival analyses need to correct for at least ancestry, sex, etc (ex. a Cox model or a stratified model, given we already know HSD17B7 will confound with ancestry).

4. Figure 3K, all the SNP's correlation have the same color? Are those SNPs always on the same haplotype, if not, what are the actual R2 and D? Great they are able to trace it to ancestry-related SNPs as eQTLs.

Language

1. Take out "co-segregating" which may suggest it's a familial linkage study)
2. "An attractive possibility..." sounds odd, consider replacing with "We postulate that..."

Referee #2 (Comments on Novelty/Model System for Author):

The authors have done a good job in developing these models, as this question has not been tested functionally to my knowledge in previous studies.

I do have some reservations about the statistics of the eQTL analysis, and would suggest that this be reviewed by a specialist in the field. However I also believe that if this was excluded it would not seriously reduce the impact of the manuscript.

Referee #2 (Remarks for Author):

The manuscript from Xu et al addresses an important question related to the differences in tumor susceptibility driven by inherited polymorphic genetic variants linked to race/ethnicity. This is a very complex problem involving both genetic and environmental factors, compounded by socio-economic differences and other influences that have been very difficult to disentangle to identify underlying mechanisms. The authors have taken a direct route to this question by studying the effect of ancestry on properties associated with transformation and oncogenicity using keratinocyte cells derived from young male foreskins. Remarkably, they identify HSD17B7 as a polymorphic gene carrying a heritable polymorphism that in the black ancestry patients is linked to increased clonal growth, reduced differentiation and oxidative phosphorylation, and increased tumorigenic potential. The authors further show that manipulation of HSD17B7 levels alters these properties in human keratinocyte cultures, and suggest that HSD17B7 is a target for intervention to prevent the development of cancer in human populations.

Overall, there is a lot of work presented in this manuscript, and it will be of interest to the field of cancer susceptibility and prevention. There are however some points that should be clarified by the authors.

1. There appears to be an important error in the statement on p 11, which states:

" Testing 14 HKC strains of Black versus White individuals for mitochondrial electron transfer chain (ETC) activity showed consistently higher levels in cells of the former group, accompanied by higher ATP production and mitochondrial ROS levels (Figure 6A)."

Figure 6 in fact shows that ETC activity and ATP levels are higher in the latter (white) group rather than the former (black) group.

2. The analysis of Fig 2 states: "Out of a hallmark of 50 gene signatures from the Molecular

Signature Database (MSigDB 50 hallmarks: <https://www.gsea-msigdb.org/gsea/msigdb>), we found a significant enrichment (FDR<0.05) for a mitochondrial oxidative phosphorylation (OXPHOS) gene signature, with no significant enrichment for other signatures (Figure 2D):

Could the authors clarify the directionality of this effect. From the figure it seems that the oxidative phosphorylation signature is INCREASED in the samples from black patients, whereas later figures stress that the samples from white patients have high oxidative phosphorylation and ATP levels.

3. The gene expression and eQTL data are difficult to evaluate and do not really contribute much to the argument. In Table 3, a large number of other SNPs have a more significant correlation with Hsd17b7 expression than those highlighted, so it is difficult to know how significant these are in the absence of further functional studies involving mutagenesis of the proposed functional binding sites surrounding the HSD17B7 gene.

4. Is there independent evidence from other data sources (eg GTEX) for the presence of these eQTLs and their significance?

5. Supp Table 2 shows that the fold change in level of expression of HSD17B7 between black and white populations is 1.7 or 1.4. Is this enough? In the transfection experiments the over-expression looks like several fold higher. Do the authors believe that 1.4 -1.7 fold is sufficient to explain the effects on clonal growth and metabolism?

6. The authors used data bases from other sources to support the conclusions regarding differential gene expression in their human keratinocytes (H&N cancer, 520 patients, TCGA Firehose Legacy). Were these all males or did they include both sexes? The data generated in the present study were all from males but I did not see a statement saying that females were excluded from the TCGA samples. If this was not done, how does gender affect the results?

Referee #1 (Remarks for Author):

Xu & Tassone et al. presented an intriguing study investigating the molecular differences observed in HKCs and HNSCCs or Black African vs. White Caucasian individuals. Combining in vitro, in vivo, and computational analyses, they were able to trace key differences to the HSD17B7 gene that have ancestry-associated eQTL. The manuscript is generally well-written and results well-presented. The reviewer has a background in cancer genomics, but not in experimental biology, and will focus on commenting on those sections.

Major:

1. As the author suggested, they "will be employing the term "ancestry" in reference to individuals with common genetic and phenotypic features rather than "race." The reviewer agree with this approach as race is a social construct, whereas genetic ancestry enables investigation of biological hypotheses. Given that the entire paper focused on comparing HKCs/HNSCCs across African and Caucasian ancestries, the authors should first define the African vs. Caucasian HKCs using genetic PCA (rather than "Black" and "White") before other analyses. This is a common practice in the human genetics field to avoid confounding and inaccuracy of self-reported race/ethnicity or potential samples swaps. a. Typically admixed individuals (ex. >20% of other ancestry) would be separately analyzed. It seems from Fig 2B that roughly half of the individuals the authors defined as "Black" would be admixed. After first defining the ancestry, the author could either keep the two groups but define the African-ancestry group as "individuals with >x% African ancestry", or more appropriately, separate the African ancestry, admixed, and Caucasian individuals.

Answer: We thank the reviewer for the appreciation of our work and the constructive recommendations. As requested, we have moved the genotyping and population admixture analysis to the beginning of the results section, and subdivided our subsequent analysis of primary keratinocytes (HKCs) from Black individuals on the basis of < or > 20% genome admixture.

As now stated at the beginning of the results (p. 5, line 13) and shown in the new Fig 1, "principal component analysis (PCA) of the SNP profiles showed excellent correspondence of genetic profiles with skin phototypes and patients' "self-reported" origins. Samples from individuals of White descent and skin phototypes 1 and 2 clustered tightly in one PCA group, while those of Black descent and skin phototypes 5 and 6 clustered separately, with three sub-groups matching geographic distributions of "self-reported" origin within the African continent (Fig 1A). Admixture analysis of the SNP genotype dataset was used to estimate genetic relatedness of donors, clustering them according to an increasing number of possible ancestral populations. With the simplest assumption of two ancestries ($K = 2$), the genome of individuals of self-reported Black origin was found to harbor various levels of the White ancestry genome (Fig 1B). The findings are consistent with the greater genetic variation of Black African populations as we consider in the discussion (p. 15, line 17).

In term of functional analysis, we now show that there are statistically significant differences in oncogenic potential of HKCs from White and Black individuals, considering the latter either as a total group or one with <20% genomic admixture (Fig 2A). Differences in clonogenicity and sphere forming capability between HKCs from White versus Black individuals, as a total group or one with < or > 20% genomic admixture, were all statistically significant (Fig 2E, F).

The transcriptomic profiles of HKC strains were already ordered according to the results of genome admixture analysis (Fig 3A). We have now also performed separate GO analysis of the profiles of HKCs from White versus Black individuals, considered as either a total group or one with <20% genomic admixture, finding similar or identical functional categories of genes in the two cases (Fig 3B).

2. *Why did the author choose HNSC for the human cohort analyses? Are there datasets of other keratinocyte cancers (especially those affecting the skin?)*

Answer : As indicated in the text (p. 8, line 13), we compared transcriptomic profiles of HKCs from Black versus White individuals with those of a large data set of Head and Neck SCCs (HNSCCs) from patients of the two ancestries (520 patients, 452 White, 48 Black; TCGA Firehose Legacy, November 2020, in cBioportal (Gao et al., 2013)). Transcriptomic profiles of esophageal and lung SCCs could not be similarly analyzed as only a few were from patients of Black descent (5 out of 114 esophageal SCCs; 31 out of 351 lung SCCs). Large skin SCC datasets with ancestry/survival information are not available, in either TCGA or other repositories (e.g. GEO, ArrayExpress).

3. *Figure 3C, the survival analyses need to correct for at least ancestry, sex, etc (ex. a Cox model or a stratified model, given we already know HSD17B7 will confound with ancestry).*

Answer: As recommended, we have looked into these other determinants of cancer susceptibility. Kaplan Mayer curves of patients' survival divided by sex show HSD17B7-dependent differences for both male and female patients (revised Fig 4B). As a second approach, multiple-variable Cox regression analysis was used to adjust for patients' sex, age and ancestry, or all three variables together, showing that even in these cases elevated HSD17B7 levels remain significantly associated with poor patients' survival (Fig 4C).

4. *Figure 3K, all the SNP's correlation have the same color? Are those SNPs always on the same haplotype, if not, what are the actual R2 and D? Great they are able to trace it to ancestry-related SNPs as eQTLs.*

Answer: We had used single Red and Blue colors for R2 and D' values, respectively, as both are close to upper limits. In revised Fig 4K, we are now showing shaded levels of color corresponding to the indicated gradient of values. The specific R2 and D' numbers are provided in Appendix Table S3.

Language

1. *Take out "co-segregating" which may suggest it's a familial linkage study)*

2. *"An attractive possibility..." sounds odd, consider replacing with "We postulate that..."*

Answer : We have removed the "co-segregating" from the text as recommended, and replaced "an attractive possibility" with "We postulate that..." (p5, line5).

Referee #2 (Comments on Novelty/Model System for Author):

Referee #2 (Remarks for Author):

The manuscript from Xu et al addresses an important question related to the differences in tumor susceptibility driven by inherited polymorphic genetic variants linked to race/ethnicity. This is a very complex problem involving both genetic and environmental factors, compounded by socio-economic differences and other influences that have been very difficult to disentangle to identify underlying mechanisms. The authors have taken a direct route to this question by studying the effect of ancestry on properties associated with transformation and oncogenicity using keratinocyte cells derived from young male foreskins. Remarkably, they identify HSD17B7 as a polymorphic gene carrying a heritable polymorphism that in the black ancestry patients is linked to increased clonal growth, reduced differentiation and oxidative phosphorylation, and increased tumorigenic potential. The authors further show that manipulation of HSD17B7 levels alters these properties in human keratinocyte cultures, and suggest that HSD17B7 is a target for intervention to prevent the development of cancer in human populations.

Overall, there is a lot of work presented in this manuscript, and it will be of interest to the field of cancer susceptibility and prevention. There are however some points that should be clarified by the authors.

1. There appears to be an important error in the statement on p 11, which states: " Testing 14 HKC strains of Black versus White individuals for mitochondrial electron transfer chain (ETC) activity showed consistently higher levels in cells of the former group, accompanied by higher ATP production and mitochondrial ROS levels (Figure 6A)."

Figure 6 in fact shows that ETC activity and ATP levels are higher in the latter (white) group rather than the former (black) group.

Answer: We thank the reviewer for pointing out the wrong wording in the text, which we have now rectified.

2. The analysis of Fig 2 states: "Out of a hallmark of 50 gene signatures from the Molecular Signature Database (MSigDB 50 hallmarks: <https://www.gsea-msigdb.org/gsea/msigdb>), we found a significant enrichment (FDR<0.05) for a mitochondrial oxidative phosphorylation (OXPHOS) gene signature, with no significant enrichment for other signatures (Figure 2D)."

Could the authors clarify the directionality of this effect. From the figure it seems that the oxidative phosphorylation signature is INCREASED in the samples from black patients, whereas later figures stress that the samples from white patients have high oxidative phosphorylation and ATP levels.

Answer : We thank the reviewer for the interesting question, which we have specifically addressed by additional data that we had obtained since submission of our manuscript. As we now indicate in the text (p.12, line 6), and show in the new Fig 7: while transcriptomic profiles of Black African versus Caucasian HKCs were distinguished by an OXPHOS related gene signature (Fig 3C). Many genes of the signature related to cellular respiration, electron transport chain and mitochondrial organization and biogenesis were more highly expressed in Black vs White HKCs and up-regulated in three strains of White HKCs by *HSD17B7* overexpression (Fig 7 and Appendix Table S1). Expression of the above genes may be inversely related to intrinsic levels of mitochondrial activity as possible compensatory mechanism as reported for mitochondria disorders and metabolic conditions resulting in

OXPHOS deficiency {Reinecke, 2009 #15586; Singh, 2020 #15585}. In fact, direct analysis of HKC strains of Black versus White individuals showed consistently lower levels of mitochondrial activity (Fig 8A), which were also reduced as a consequence of *HSD17B7* overexpression (Fig 8B,C).

3. The gene expression and eQTL data are difficult to evaluate and do not really contribute much to the argument. In Table 3, a large number of other SNPs have a more significant correlation with Hsd17b7 expression than those highlighted, so it is difficult to know how significant these are in the absence of further functional studies involving mutagenesis of the proposed functional binding sites surrounding the HSD17B7 gene.

Answer : In Appendix Table S3 we had included the results of significant cis-eQTLs for all the differentially expressed genes (164) found in Black versus White HKCs. Only 9 eQTLs were found to be associated with the *HSD17B7* locus (1 MB either side of the gene). This is more clearly indicated in title of Appendix Table S3 and, to avoid any confusion, we have added a second work sheet to the Appendix Table S3, listing only the 9 HSD17B7-associated eQTLs together with their ancestry distribution (Shown by Fst values). We have also added to the Appendix Table S3 a worksheet with the results of co-segregation analysis for the HSD17B7 e-QTLs, as shown in a graphic form in Fig 4K.

4. Is there independent evidence from other data sources (eg GTEX) for the presence of these eQTLs and their significance?

Answer : Yes, we looked at the 6 ancestry specific eQTLs for *HSD17B7* in the GTEX database and found that all of them are significantly associated with *HSD17B7* gene expression in multiple tissues including skin and surface epithelia (esophagus mucosa). The findings are now mentioned in the text (p. 10) and shown in Appendix Table S4.

5. Supp Table 2 shows that the fold change in level of expression of HSD17B7 between black and white populations is 1.7 or 1.4. Is this enough? In the transfection experiments the over-expression looks like several fold higher. Do the authors believe that 1.4 -1.7 fold is sufficient to explain the effects on clonal growth and metabolism?

Answer : 1.7 and 1.4 fold differences in *HSD17B7* expression levels are average values based on comparative analysis of all HKC strains and HNSCCs from the TCGA database, with variations in expression of the gene across samples significantly correlating with patients' survival and, in the case of HKCs, OXPHOS gene signature and clonogenic potential.

It would be very difficult if not impossible to reproduce the relative slight differences in levels of *HSD17B7* expression between HKCs of the two ancestries by lentiviral-mediated overexpression or silencing. We note that even the elevated lentiviral-mediated overexpression of *HSD17B7* enhanced stem cell potential of a number of HKC strains (of white origin) but not others, consistent with an interplay with other determinants of stem cell potential. More drastic effects were observed by the gene silencing approach across HKC strains of the two ancestries, unveiling its essential function in this cell type.

Overall, as we summarize at the beginning of the discussion (p. 15), "our combined evidence, stemming from analysis of keratinocytes and keratinocyte-derived tumors from individuals of Black African versus Caucasian ancestries, has led to a differentially expressed gene of unsuspected importance in control of keratinocyte stem cell and oncogenic potential as well mitochondrial OXPHOS activity This gene is a likely co-determinant of the

observed differences between keratinocytes of the two ancestries in concert with other as yet to be determined factors”.

6. The authors used data bases from other sources to support the conclusions regarding differential gene expression in their human keratinocytes (H&N cancer, 520 patients, TCGA Firehose Legacy). Were these all males or did they include both sexes? The data generated in the present study were all from males but I did not see a statement saying that females were excluded from the TCGA samples. If this was not done, how does gender affect the results?

Answer : We thank the reviewer for the interesting question. As already indicated in reply to reviewer #1, we have looked into these and other determinants of cancer susceptibility. Kaplan Mayer curves of patients’ survival divided by sex show *HSD17B7*-dependent differences for both male and female patients (revised Fig 4B). As a second approach, multiple-variable Cox regression analysis was used to adjust for patients’ sex, age and ancestry, or all three variables together, showing that even in these cases elevated *HSD17B7* levels remain significantly associated with poor patients’ survival (Fig 4C).

5th May 2021

Dear Paolo,

Thank you for the submission of your revised manuscript to EMBO Molecular Medicine. We have now received the enclosed report from the referee who re-reviewed your manuscript. As you will see, this referee is supportive of publication, and I am therefore pleased to inform you that we will be able to accept your manuscript once the following editorial points will be addressed:

1/ Main manuscript text:

- Please answer/correct the changes suggested by our data editors in the main manuscript file (in track changes mode). This file will be sent to you in the next couple of days. Please use this file for any further modification.
- Please provide up to 5 keywords.
- Material and methods:
 - o Human samples: please include a statement that written informed consent was obtained from all subjects and that the experiments conformed to the principles set out in the WMA Declaration of Helsinki and the Department of Health and Human Services Belmont Report.
 - o Cells: please indicate the origin of the cells.
 - o Animals: please indicate the origin of the mice.
 - o Antibodies: please provide antibody dilutions.
- Thank you for providing a Data Availability section. Please note that data have to be made public before acceptance of the manuscript.

2/ Figures:

- Statistics: Please indicate in all main and appendix figures (or in their legends) the exact p= values (including non-significant p values, ns). You may provide these values as a supplemental table in the Appendix file.
- Please make sure that all figures/figure panels are referenced in the main text and in the chronological order in which the figures appear (callouts are missing for Fig 7A; Fig 7B is called out after Fig 9).
- The Appendix Tables S1, 3, 5 should be renamed "Dataset EV1, 2, 3".
- Appendix Tables S2 and 3 should be renamed "Table EV1 and 2".
- All appendix figures/tables and EV tables need their legends removed from the main manuscript file and added to the respective files.
- Appendix: please add a table of content.

3/ Checklist:

- Section D/8: please provide the housing and husbandry conditions of the mice.
- Section E/12: please include a statement that written informed consent was obtained from all subjects and that the experiments conformed to the principles set out in the WMA Declaration of Helsinki and the Department of Health and Human Services Belmont Report.

4/ Source Data: Thank you for providing raw data for Figure 6. Please upload them as one pdf file.

5/ The paper explained: EMBO Molecular Medicine articles are accompanied by a summary of the

articles to emphasize the major findings in the paper and their medical implications for the non-specialist reader. Please provide a draft summary of your article highlighting

6/ Synopsis: I slightly edited your synopsis to fit our style and format, please let me know if you agree with the following:

Differences in individuals' cancer susceptibility can be attributed, in part, to specific genetic and epigenetic variations. Human populations of Black African ancestry have a higher risk of aggressive cancer of various types, including keratinocyte-derived squamous cell carcinomas (SCCs).

- Higher oncogenic and self-renewal potential with lower mitochondrial respiratory and OXPHOS activities were observed in keratinocytes from Black African versus White Caucasian individuals.
- HSD17B7 was the top-ranked differentially expressed gene in primary keratinocytes and Head/Neck SCCs from Black African versus Caucasian populations, with ancestry-specific eQTLs linked to its expression.
- HSD17B7 codes for a targetable enzyme involved in sex steroid and cholesterol biosynthesis.
- HSD17B7 was found to play a key role in control of keratinocyte stem cell and oncogenic potential as well as mitochondrial OXPHOS activity.

7/ As part of the EMBO Publications transparent editorial process initiative (see our Editorial at <http://embomolmed.embopress.org/content/2/9/329>), EMBO Molecular Medicine will publish online a Review Process File (RPF) to accompany accepted manuscripts.

This file will be published in conjunction with your paper and will include the anonymous referee reports, your point-by-point response and all pertinent correspondence relating to the manuscript. Let us know whether you agree with the publication of the RPF and as here, if you want to remove or not any figures from it prior to publication.

I look forward to receiving your revised manuscript.

With my best wishes,

Lise

Lise Roth, PhD
Editor
EMBO Molecular Medicine

To submit your manuscript, please follow this link:

Link Not Available

Photos 400-800 DPI

*Additional important information regarding figures and illustrations can be found at <https://bit.ly/EMBOPressFigurePreparationGuideline>

The system will prompt you to fill in your funding and payment information. This will allow Wiley to send you a quote for the article processing charge (APC) in case of acceptance. This quote takes into account any reduction or fee waivers that you may be eligible for. Authors do not need to pay any fees before their manuscript is accepted and transferred to our publisher.

***** Reviewer's comments *****

Referee #1 (Remarks for Author):

I am satisfied with the revised manuscript. The authors are to be congratulated on this impressive body of work combining a new cohort, genomics, and biochemical experiment.

The authors performed the requested editorial changes.

20th May 2021

Dear Paolo,

I am very pleased to inform you that your manuscript is now accepted for publication and will be sent to our publisher to be included in the next available issue of EMBO Molecular Medicine!

We would like to remind you that as part of the EMBO Publications transparent editorial process initiative, EMBO Molecular Medicine will publish a Review Process File online to accompany accepted manuscripts. If you do NOT want the file to be published or would like to exclude figures, please immediately inform the editorial office via e-mail.

Congratulations on your interesting work!

With my best wishes,

Lise

Lise Roth, Ph.D
Editor
EMBO Molecular Medicine

Follow us on Twitter @EmboMolMed
Sign up for eTOCs at embopress.org/alertsfeeds

YOU MUST COMPLETE ALL CELLS WITH A PINK BACKGROUND ↓
PLEASE NOTE THAT THIS CHECKLIST WILL BE PUBLISHED ALONGSIDE YOUR PAPER

Corresponding Author Name: Gian Paolo Dotto

Manuscript Number: EMM-2021-14133-V2